# c-Myc plays a key role in IFN-γ-induced persistence of *Chlamydia trachomatis*

Nadine Vollmuth[1], Lisa Schlicker[2], Yongxia Guo[1,3], Pargev Hovhannisyan[1], Sudha Janaki-Raman[4], Naziia Kurmasheva[1], Werner Schmitz[5], Almut Schulze[2,5], Kathrin Stelzner[1], Karthika Rajeeve[1,6†], Thomas Rudel[1]*†

[1]Department of Microbiology, Biocenter, University of Wuerzburg, Wuerzburg, Germany; [2]German Cancer Research Center (DKFZ), Heidelberg, Germany; [3]College of Veterinary Medicine, China Agricultural University, Beijing, China; [4]Memorial Sloan Kettering Cancer Center, New York, United States; [5]Department of Biochemistry and Molecular Biology, University of Wuerzburg, Würzburg, Germany; [6]Pathogen Biology, Rajiv Gandhi Centre for Biotechnology (RGCB), Thiruvananthapuram, India

**Abstract** *Chlamydia trachomatis (Ctr)* can persist over extended times within their host cell and thereby establish chronic infections. One of the major inducers of chlamydial persistence is interferon-gamma (IFN-γ) released by immune cells as a mechanism of immune defence. IFN-γ activates the catabolic depletion of L-tryptophan (Trp) via indoleamine-2,3-dioxygenase (IDO), resulting in persistent *Ctr*. Here, we show that IFN-γ induces the downregulation of c-Myc, the key regulator of host cell metabolism, in a STAT1-dependent manner. Expression of c-Myc rescued *Ctr* from IFN-γ-induced persistence in cell lines and human fallopian tube organoids. Trp concentrations control c-Myc levels most likely via the PI3K-GSK3β axis. Unbiased metabolic analysis revealed that *Ctr* infection reprograms the host cell tricarboxylic acid (TCA) cycle to support pyrimidine biosynthesis. Addition of TCA cycle intermediates or pyrimidine/purine nucleosides to infected cells rescued *Ctr* from IFN-γ-induced persistence. Thus, our results challenge the longstanding hypothesis of Trp depletion through IDO as the major mechanism of IFN-γ-induced metabolic immune defence and significantly extends the understanding of the role of IFN-γ as a broad modulator of host cell metabolism.

*For correspondence:
thomas.rudel@uni-wuerzburg.de

†These authors contributed equally to this work

Competing interest: The authors declare that no competing interests exist.

## Editor's evaluation

This paper will be of interest to scientists working to understand *Chlamydia trachomatis* persistence, and host pathogen interaction in general. The authors report the surprising observation that the mechanism of restriction of bacterial growth is through the inhibition of c-Myc signaling by IFNγ as opposed to IDO-dependent depletion of tryptophan levels, as had been previously suggested.

## Introduction

*Chlamydia trachomatis (Ctr)* is an obligate intracellular human pathogen, which causes a broad range of acute and chronic diseases (*Fenwick, 2012*). It is the leading cause of bacterial sexually transmitted diseases (STD), with more than 130 million new cases annually (*Newman et al., 2015*). Infection of the urogenital tract by *Chlamydia* can lead to urethritis, infertility, ectopic pregnancies, and pelvic inflammatory disease (PID) (*Malik et al., 2009*; *Svenstrup et al., 2008*). Furthermore, left untreated, *Chlamydia* infection increases the risk of HIV infections (*Galvin and Cohen, 2004*; *Ho et al., 1995*) and might contribute to the development of cervical and ovarian cancer (*Gagnaire et al., 2017*; *Hare et al., 1982*; *Koskela et al., 2000*; *Smith et al., 2004*). The pathogen extensively interferes with the

physiology of the infected cell, since it depends entirely on its host cell as a replicative niche during infection (*Wang et al., 2011*).

*Chlamydiae* have a unique biphasic life cycle, consisting of two physiologically and morphologically distinct forms (*Gaylord, 1954*). This gram-negative bacterium initiates its developmental cycle by attachment and invasion of the host cell by the elementary body (EB), the non-dividing infectious form of the pathogen. Inside the cell, EB stay within the endosomes, which they modify rapidly to create a replicative niche termed 'inclusion', to avoid lysosomal degradation (*Hackstadt et al., 1997*). EB differentiate into reticulate bodies (RB), which is the non-infectious replicating form of the pathogen. It takes several rounds of cell division until RB re-differentiate back into EB and the progenies are released by host cell lysis or extrusion, an exocytosis like mechanism, to infect neighbouring cells (*Abdelrahman and Belland, 2005*; *Hybiske and Stephens, 2007*). Apart from active infection, *Chlamydia* can turn into a dormant state called persistence and can remain over a long time, maybe even years (*Suchland et al., 2017*) within its host cell and thereby establish a chronic infection. During the persistent state, the pathogen remains viable and replicates its genome, but it exhibits decreased metabolic activity and inhibited cell division, which leads to the formation of enlarged pleomorphic aberrant bodies (AB) (*Wyrick, 2010*). The transition to the persistent form might represent an important chlamydial survival mechanism against antibiotics and the immune response of the host, since this process is reversible (*Beatty et al., 1994*; *Wyrick, 2010*). The conversion to this dormant stage is induced by penicillin (*Tamura and Manire, 1968*), iron deficiency (*Raulston, 1997*), amino acid starvation (*Allan and Pearce, 1983b*), and interferon-gamma (IFN-γ) (*Beatty et al., 1994*; *Wyrick, 2010*). IFN-γ is an immune-regulated cytokine, which is involved in the cell-intrinsic immunity against several intracellular pathogens, including *Chlamydia* (*MacMicking, 2012*). The downstream signalling pathways activated by IFN-γ are highly species specific and differ dramatically between mouse and human cells (*Nelson et al., 2005*). In human cells, the anti-chlamydial effect of IFN-γ is predominantly mediated by the induction of indoleamine-2,3-dioxygenase (IDO), an enzyme that catalyses the initial step of L-tryptophan (Trp) degradation to *N*-formyl kynurenine and kynurenine (*Taylor and Feng, 1991*). *Ctr* as a Trp auxotroph needs to take up Trp from the host cell for its development (*Østergaard et al., 2016*; *Wyrick, 2010*). Trp depletion by IFN-γ treatment therefore leads to persistence (*Beatty et al., 1994*; *Byrne et al., 1986*). However, *Ctr* contain a tryptophan synthase operon comprised of genes encoding the tryptophan repressor (TrpR) and tryptophan synthase α and β subunits (TrpA and TrpB, respectively) (*Fehlner-Gardiner et al., 2002*). *Chlamydia* with an intact tryptophan synthase operon have been shown to produce Trp from indole, a metabolic activity that can rescue these strains from IFN-γ-induced growth restriction, if indole is available at the site of infection (*Caldwell et al., 2003*; *Ziklo et al., 2016*).

IFN-γ belongs to the type II interferons that bind to the extracellular domain of the IFN-γ receptor, which is a heterodimer of the two subunits IFNGR1 and IFNGR2. The intracellular domains of the IFNGR1 subunits are associated with Janus kinase 1 (Jak1), while the IFNGR2 subunits are associated with Jak2. Activation of Jak1 and Jak2 results in phosphorylation of the receptor and subsequent recruitment and phosphorylation of signal transducer and activator of transcription (STAT1). STAT1 phosphorylation at tyrosine 701 and serine 727 leads to its homodimerization and nuclear translocation. Once in the nucleus, STAT1 homodimers bind to IFN-γ-activated sequence (GAS) elements in the promoters of target genes to regulate their transcription (*Hu and Ivashkiv, 2009*; *Krause et al., 2006*; *Ramana et al., 2000*). IFN-γ can function as both a growth inhibiting and promoting cytokine in a STAT1-dependent manner (*Asao and Fu, 2000*; *Ramana et al., 2000*). Binding of STAT1 homodimers to the consensus GAS elements in the *c-myc* promoter inhibits its expression transcriptionally (*Ramana et al., 2000*). Concurrently, *c-myc* expression is not only negatively regulated by STAT1 but *stat1* is also a negative target gene of c-Myc. Hence, c-Myc and STAT1 regulate each other in a negative feedback loop at the transcriptional level (*Schlee et al., 2007*).

The transcription factor c-Myc targets genes involved in the regulation of numerous cellular processes, such as cell proliferation, cell growth, translation, metabolism, and apoptosis (*Battey et al., 1983*; *Coffin et al., 1981*; *Dang, 1999*; *Duesberg and Vogt, 1979*). Furthermore, c-Myc activity increases energy production, anabolic metabolism, promotion of aerobic glycolysis, and glutaminolysis, inducing mitochondrial biogenesis and tricarboxylic acid (TCA) cycle activity (*Dang, 1999*). Glutaminolysis increases the production of biomass by providing TCA cycle intermediates via anaplerosis, which enhances their availability for the production of amino acids, nucleotides, and lipids (*Kress*

*et al., 2015*). c-Myc also critically controls nucleotide biosynthesis by directly regulating the expression of genes that encode the enzymes involved in the production of precursors of all nucleotides (*Liu et al., 2008*; *Mannava et al., 2008*). For example, c-Myc directly controls the cis-regulatory element in the 5'-UTR of phosphoribosyl pyrophosphate synthetase, which catalyses the first committed step in purine biosynthesis (*Cunningham et al., 2014*). In pyrimidine biosynthesis, the rate-limiting step is catalysed by carbamoyl aspartate dehydratase (CAD) (*Evans and Guy, 2004*), which is also regulated by c-Myc as a response to growth stimulatory signals, such as activation of EGFR/RAS/MAP kinase (*Makinoshima et al., 2014*), Hif 1/2 α (*Gordan et al., 2007*), or estrogen receptor/Sp1 (*Khan et al., 2003*) pathways. We recently identified a central role of c-Myc in the control of the metabolism in *Chlamydia*-infected cells (*Rajeeve et al., 2020*). *Chlamydia* has a reduced genome and only very limited metabolic capacity. For example, they have a truncated TCA cycle and lack the ability to synthesize purine and pyrimidine nucleotides de novo but acquire ATP and nucleosides from the host cell (*McClarty and Tipples, 1991*; *Tipples and McClarty, 1993*).

Expression of IDO and depletion of Trp has been previously described as the main reason for interferon-induced persistence in *Chlamydia* (*Aiyar et al., 2014*; *Panzetta et al., 2018*). Here, we show that IFN-γ induces a STAT1-dependent depletion of c-Myc in *Chlamydia*-infected cells, which de-regulates the host metabolism and induces *Chlamydia* persistence. Importantly, addition of the TCA cycle intermediate α-ketoglutarate or pyrimidine/purine nucleosides were sufficient to prevent persistence and restore chlamydial replication. Our data demonstrate a central role of c-Myc-regulated metabolic pathways in the IFN-γ-induced persistence of *Ctr*.

## Results

### IFN-γ treatment prevents c-Myc induction and impairs the development of *Chlamydia*

c-Myc is an important regulator of host cell metabolism and is indispensable for chlamydial acute infection and progeny formation (*Rajeeve et al., 2020*). Since *Chlamydia* adapts a parasitic lifestyle, and it has been shown that several environmental conditions affecting host cell metabolism like iron and amino acid starvation also induce chlamydial persistence (*Allan and Pearce, 1983a*; *Raulston, 1997*), we investigated the role of c-Myc in chlamydial persistence. From our previous study, it was known that ablation of c-Myc expression interfered with chlamydial development and progeny formation (*Rajeeve et al., 2020*). To investigate if *Chlamydia* enter a persistence state in c-Myc-depleted cells, we infected a HeLa 229 cell line with an anhydrotetracycline (AHT)-inducible short hairpin RNA (shRNA) for c-Myc in the absence and presence of the inducer. In agreement with our previous results (*Rajeeve et al., 2020*), silencing of c-Myc expression prevented normal inclusion formation and chlamydial development (*Figure 1A, B and C*). We then cultivated cells for 24 hr with AHT to suppress c-Myc expression (conditions 3 and 4; see *Figure 1A*). At this 24 hr time point, infection load and c-Myc levels were similarly low for conditions 3 and 4. We then either added AHT for another 12 hr (condition 3) or removed AHT (condition 4) to re-establish c-Myc expression (see scheme *Figure 1A*). In contrast to the cells with silenced c-Myc, removal of AHT efficiently restored inclusion formation (*Figure 1A, B and C*), suggesting that suppression of c-Myc induces a persistence state in *Chlamydia*. The bacteria obtained in the conditions 2, 3, and 4 were replated on fresh HeLa cell to test if they formed infectious EB under these conditions. These so-called secondary infections demonstrated that the development of EB was severely affected in condition 3, but not in the conditions 2 and 4 (*Figure 1B and C*). AHT alone did not impact inclusion formation in the absence of c-Myc suppression (*Figure 1—figure supplement 1A, B*).

A physiological mechanism to induce chlamydial persistence is the exposure of infected cells to IFN-γ which causes the depletion of Trp via the induction of IDO (*Taylor and Feng, 1991*). We therefore investigated the connection of persistence induced by IFN-γ and c-Myc depletion. Interestingly, IFN-γ-treated cells failed to stabilize c-Myc upon *Chlamydia* infection (*Figure 1D*). In contrast, induction of persistence by antibiotic treatment, which targets the bacterial rather than host cell metabolism, did not alter c-Myc levels (*Figure 1D*). Nevertheless, both modes of persistence induction caused a failure of the bacteria to produce infectious progenies (*Figure 1—figure supplement 1C, D*). To test whether silencing of c-Myc also induces depletion of Trp in *Ctr*, we monitored the transcription of the chlamydial Trp synthase gene *trp*B which is strongly induced upon endogenous Trp depletion by IFN-γ

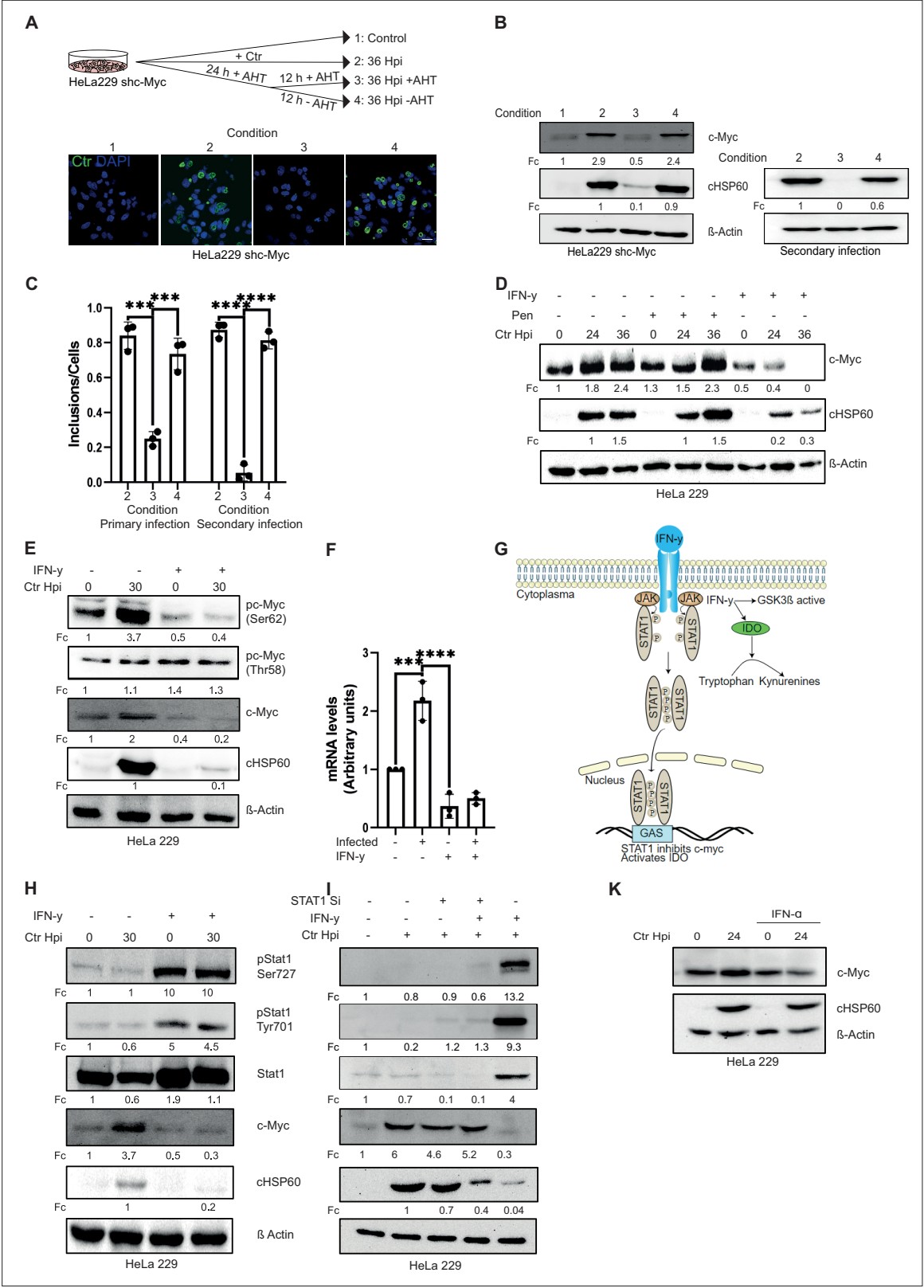

**Figure 1.** Interferon-gamma (IFN-γ) induces depletion of c-Myc and impairs chlamydial growth. (**A**) HeLa 229 cells with an anhydrotetracycline (AHT)-inducible expression of shc-Myc were infected with *Chlamydia trachomatis* (*Ctr*) at multiplicity of infection (MOI) 1. The infected cells were either left untreated or treated with 100 ng/mL AHT to deplete c-Myc 8 hr before infection. After 24 hr of infection, AHT was removed to release c-Myc expression, and the restoration of inclusion formation was tested. Cells were either fixed with 4% PFA after 36 hpi and immunostained for *Ctr* (cHSP60: green) and

*Figure 1 continued on next page*

*Figure 1 continued*

DNA (DAPI: blue) or lysed and analysed by (**B**) Western blot in order to analyse the rescue. Additionally, infected cells were lysed to infect freshly plated HeLa 229 shc-Myc cells at 24 hpi and analysed via Western blot to investigate the formation of infectious progenies. The panel shows the image of one representative blot (n=3). cHSP60 indicates *Chlamydia* infection and β-Actin staining serves as the loading control (n=3). (**C**) Inclusions and cells were counted and the results are shown as bar diagram. One-way ANOVA was used for analysis. *** indicates p-value <0.001, **** indicates a p-value <0.0001. (**D**) HeLa 229 cells were either left untreated or were pre-treated for 2 hr with 10 ng/mL IFN-γ or 1 unit of penicillin, infected with *Ctr* at MOI 1 for different time intervals and lysed for Western blot analysis. Bacterial load (cHSP60) and c-Myc levels were determined, and β-Actin served as loading control (n=3). (**E**) HeLa 229 cells were either left untreated or were pre-treated for 2 hr with 10 ng/mL of IFN-γ and infected with *Chlamydia* (*Ctr*) at MOI 1 and lysed at 30 hr to perform Western blot analysis. Phosphorylated form of c-Myc at serine 62 (pc-Myc [Ser62]) and threonine 58 (pc-Myc [Thr58]), c-Myc, and *Chlamydia* (cHSP60) were detected and quantified (Fc) (n=3). (**F**) HeLa 229 cells were either left untreated or treated with 10 ng/mL of IFN-γ and infected with *Ctr*. The cells were lysed and relative mRNA levels of c-Myc were determined by qPCR. GAPDH was used for normalization (n=3). One-way ANOVA was used for analysis. *** indicates p-value <0.001, **** indicates a p-value <0.0001. (**G**) Cartoon depicting IFN-γ signalling. IFN-γ binds to the IFN-γ receptor which results in the phosphorylation of STAT1. pSTAT1 binds to the IFN-γ-activated sequence (GAS) sequence and thus blocks c-Myc transcription. IFN-γ can also induce indoleamine-2,3-dioxygenase (IDO) and thereby the degradation of L-tryptophan. (**H**) HeLa 229 cells were either left untreated or were pre-treated for 2 hr with 10 ng/mL of IFN-γ and infected with *Chlamydia* (*Ctr*) at MOI 1 and lysed at 30 hpi to study STAT1 signalling after *Ctr* infection (n=3). (**I**) HeLa 229 cells were transfected with siRNA against STAT1 (+) or control (-) for 48 hr and then infected with *Ctr* for 24 hr. The cells were lysed and analysed by Western blot to investigate the STAT1 signalling and infectivity of *Ctr* (n=3). (**J**) HeLa 229 cells were either left untreated or were pre-treated for 2 hr with 10 ng/mL of IFN-α and infected with *Ctr* at MOI 1 and lysed at 36 hpi and analysed via Western blot. For all Western blots shown in **A-J**, *Chlamydia* load (cHSP60) and the respective host cell protein levels were quantified by normalization to β-Actin and indicated as fold change (Fc). Image of Western blots was taken from one of a total of at least three blots of biological replicates.

The online version of this article includes the following source data and figure supplement(s) for figure 1:

**Source data 1.** Complete and cutted membranes of all Western blots from *Figure 1*.

**Figure supplement 1.** Interferon-gamma (IFN-γ) induces c-Myc depletion and impairs chlamydial growth.

**Figure supplement 1—source data 1.** Complete and cutted membranes of all Western blots from *Figure 1—figure supplement 1*.

treatment (*Figure 1—figure supplement 1K*). However, silencing of c-Myc did not cause significant changes in *trp*B transcription (*Figure 1—figure supplement 1L*), indicating that c-Myc depletion does not affect endogenous Trp levels in *Ctr*.

c-Myc protein stability is regulated by two phosphorylation sites with opposing functions. Serine 62 phosphorylation (pS62) stabilizes c-Myc whereas threonine 58 phosphorylation (pT58) promotes c-Myc degradation (*Vervoorts et al., 2006*). IFN-γ treatment led to decreased phosphorylation of c-Myc at serine 62, a modification that prevents the ubiquitination and proteasomal degradation of the protein (*Welcker et al., 2004*) while phosphorylation at threonine 58 was unchanged (*Figure 1E*). These results were obtained, if protein bands of the Western blot were normalized to actin as loading control. However, since c-Myc levels varied significantly upon IFN-γ exposure, normalization to c-Myc protein levels revealed strongly increased phosphorylation at Thr58 (4- and 7-fold, IFN-γ and IFN-γ/ infected, respectively) and a mild increase at Ser62 (1.3- and 2-fold, IFN-γ and IFN-γ/infected, respectively). These data are consistent with a role for Thr58 phosphorylation in IFN-γ-mediated downregulation of c-Myc. c-Myc was also depleted upon IFN-γ treatment of primary cells from human fimbriae (Fimb), although the concentration of IFN-γ required to achieve the same effect was five times higher than in HeLa 229 cells (10 ng/mL in HeLa 229 compared to 50 ng/mL in Fimb) (*Figure 1—figure supplement 1E, F*). Similar results were obtained with a serovar D strain involved in STD (*Figure 1—figure supplement 1G, H*).

IFN-γ is known to signal via the JAK-STAT pathway via phosphorylation of STAT1 and subsequent transcriptional regulation of gene expression (*Figure 1G*). Treatment with IFN-γ led to the phosphorylation of STAT1 at serine 727 and tyrosine 701 (*Figure 1H*) and prevented the induction of c-Myc protein (*Figure 1H*) and mRNA expression (*Figure 1F*) upon *Chlamydia* infection. To verify that interferon transcriptionally depletes c-Myc via STAT1 signalling, we used siRNA against STAT1 (*Figure 1I*). STAT1 depletion rescued c-Myc levels and chlamydial growth in primary infections (*Figure 1I*). When bacteria from IFN-γ-treated and STAT1-depleted cells were transferred to fresh cells, infectious progeny could be recovered, indicating that STAT1 downregulates c-Myc and prevents chlamydial development (*Figure 1—figure supplement 1I, J*). Since IFN-α also signals through STAT1, we tested whether IFN-α also downregulates c-Myc and chlamydial growth. Although c-Myc levels and the bacterial load were slightly reduced upon IFN-α treatment, it failed to induce strong c-Myc depletion and chlamydial persistence suggesting that type I and II IFNs have different effects on c-Myc levels and chlamydial replication (*Figure 1J*).

The IFN-γ response in humans and mice is entirely different, limiting the use of murine systems as models for human pathogenic *Ctr* infections (*Nelson et al., 2005*). We therefore established an IFN-γ-induced persistence model in human fallopian tube organoids. Healthy tissue obtained from patients that underwent hysterectomy was used to establish organoid cultures. These organoids obtained from five different patients were pre-treated with IFN-γ for 2 hr and then infected with *Chlamydia* for 6 days (*Figure 2A and B*; *Figure 2—figure supplement 1C*). In this human infection model, IFN-γ treatment strongly reduced inclusion formation (primary infection: *Figure 2B*) and production of infectious progeny (*Figure 2C and F*; *Figure 2—figure supplement 1A, B*). Moreover, the IFN-γ-STAT1 signalling axis was active in human organoids and efficiently prevented the induction of c-Myc upon infection (*Figure 2E*). When IFN-γ was removed by replacing the culture medium, a significant rescue of chlamydial inclusion formation was measured (*Figure 2D*), indicating that this system can be used as a model of persistent infection.

## Expression of c-Myc rescues *Chlamydia* from IFN-γ-induced persistence

As we observed a key role of c-Myc in chlamydial persistence, we next asked if maintaining c-Myc levels can overcome IFN-γ-induced persistence. To address this question, we used a WII-U2OS cell line, in which c-Myc expression is under the control of an AHT-inducible promoter. These cells were used as infection models for IFN-γ-induced persistence under constant c-Myc expression. Bacterial replication efficiency and the ability to produce infectious progenies were studied. Interestingly, ectopic expression of c-Myc suppressed growth inhibition induced by IFN-γ (*Figure 3A*) and supported the development of infectious progeny (*Figure 3B and C*; *Figure 3—figure supplement 1A*). Even after extended exposure of infected cells for 24 hr to IFN-γ, expression of c-Myc rescued progeny formation (*Figure 3—figure supplement 1B*). We also tested the effect of c-Myc on persistence in organoids derived from human fallopian tubes. Organoids were transduced with lentivirus expressing c-Myc or GFP as control (*Figure 3—figure supplement 1C-E*) and selected for puromycin resistance (*Figure 3—figure supplement 1F*). IFN-γ-induced c-Myc downregulation varies in different cell lines and in organoids downregulation is only visible in infected organoids due to weak c-Myc expression in uninfected human organoids (*Figure 2E*). Therefore, lentivirus-induced overexpression is also weaker in human organoids as compared to HeLa 229 cells as shown in *Figure 3—figure supplement 1C and D*. These organoids were then infected with *Chlamydia* and treated with IFN-γ. Induced expression of c-Myc rescued the strong suppression of infectious progeny development by IFN-γ also in fallopian tube organoids (*Figure 3D and E*), demonstrating that downregulation of c-Myc is essential for IFN-γ-mediated persistence in this human infection model.

## Both c-Myc and Trp are required to rescue *Chlamydia* from IFN-γ-induced persistence

It has been shown previously that IFN-γ-mediated persistence can be overcome by the addition of exogenous Trp (*Beatty et al., 1994*). To investigate the importance of Trp in the context of IFN-γ-mediated regulation of c-Myc expression in our model system, HeLa 229 and human Fimb cells were treated with IFN-γ, infected with *Chlamydia* and provided with exogenous Trp. Subsequently, bacterial replication efficiency as well as production of infectious progeny was analysed by Western blotting. As expected, addition of Trp rescued chlamydial growth after IFN-γ treatment (*Figure 4—figure supplement 1A*) and gave rise to infectious progeny (*Figure 4A*; *Figure 4—figure supplement 1B, C*). Surprisingly, addition of Trp also resulted in the stabilization of c-Myc, even without chlamydial infection (*Figure 4—figure supplement 1D*).

All *Ctr* strains are Trp auxotroph but the genital strains retain a Trp synthase (*trpB*), which uses exogenous indole provided by the microflora in the genital tract as a substrate to synthesize Trp (*Byrne et al., 1986*; *Kari et al., 2011*; *Østergaard et al., 2016*). Thus, we validated if exogenous indole recovers chlamydial growth and leads to infectious progenies. Intriguingly, indole addition resulted not only in suppression of persistence (*Figure 4B*) and formation of infectious progeny (*Figure 4—figure supplement 1E, F*), but also in the stabilization of c-Myc in both infected and non-infected cells (*Figure 4B*).

Next, we tested whether Trp alone is sufficient to support *Chlamydia* development also in the absence of c-Myc. To address this question, we depleted c-Myc by AHT-inducible shRNA-mediated gene silencing and supplemented the medium with Trp as done in previous experiments. These cells

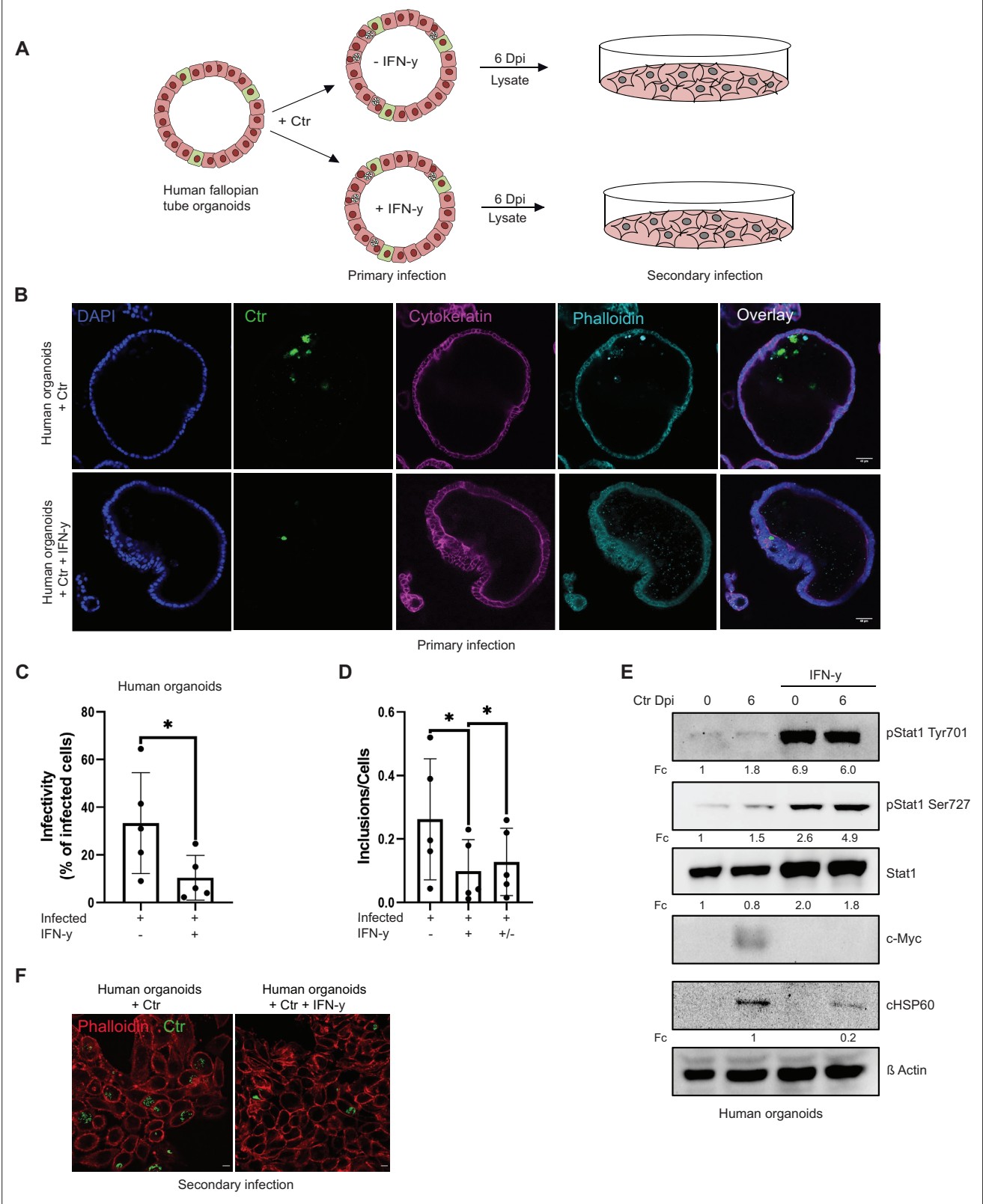

**Figure 2.** Interferon-gamma (IFN-γ) induces depletion of c-Myc and impairs chlamydial growth in human fallopian tube organoids. (**A**) Layout of the infectivity assay performed in human fallopian tube organoids. Organoids (see Materials and methods) were infected with *Chlamydia trachomatis (Ctr)* and treated with or without IFN-γ for 6 days and lysed using glass beads. Dilutions of the supernatants were used to infect freshly plated HeLa 229 cells. (**B**) Human fallopian tube organoids were infected with *Ctr* and were either left untreated or treated with IFN-γ for 6 days. The organoids were fixed

*Figure 2 continued on next page*

*Figure 2 continued*

with 4% PFA and immunostained for DNA (DAPI: blue), *Ctr* (cHSP60: green), Cytokeratin (magenta), and F-actin (Phalloidin, cyan). The panel shows representative images from organoids derived from five patients. (**C**) The infected organoids from (**B**) were lysed using glass beads, and dilutions of the supernatant were used to infect freshly plated HeLa 229 cells. The number of inclusions as shown in (**F**) were counted from five different patients and mean ± SD was depicted in the graph (n=5). * indicates p-value <0.05. (**D**) Human fallopian tube organoids were infected with *Ctr* and treated with or without IFN-γ for 6 days or IFN-γ was removed after 2 days for a 4-day recovery. Infected organoids were lysed using glass beads, and dilutions of the supernatants were used to infect freshly plated HeLa 229 cells. The number of inclusions and the number of the cells were counted to plot the graph. Paired t-test was used for analysis. * indicates a p-value <0.05. (**E**) Human fallopian tube organoids were infected with *Ctr* and treated with IFN-γ or left untreated for 6 days. The organoids were lysed in ×2 Laemmli buffer and analysed by Western blotting. *Chlamydia* load (cHSP60) and STAT1 protein levels were quantified by normalization to β-Actin and indicated as fold change (Fc). Shown are representative Western blots of at least three independent experiments (n=3).

The online version of this article includes the following source data and figure supplement(s) for figure 2:

**Source data 1.** Complete and cutted membranes of all Western blots from *Figure 2*.

**Figure supplement 1.** Interferon-gamma (IFN-γ)-induced persistence of *Chlamydia trachomatis* (*Ctr*) in human organoids.

were used for infection studies with *Chlamydia* and the formation of progeny was analysed via Western blot (*Figure 4C*, *Figure 4—figure supplement 1G*). Interestingly, the pathogen failed to develop in cells with reduced levels of c-Myc even in presence of excessive exogenous Trp (*Figure 4C*). Furthermore, *Chlamydia* was also unable to establish a secondary infection under low c-Myc expression conditions, irrespective of Trp availability (*Figure 4—figure supplement 1G*). Since excess Trp cannot overcome the suppression of c-Myc, we tested if the constant expression of c-Myc in a Trp-free environment would be able to rescue chlamydial growth. Therefore, WII-U2OS cells were cultivated in Trp-free medium, induced with AHT, treated with IFN-γ, and infected with *Chlamydia*. Bacterial replication efficiency and the ability to produce infectious progeny was examined by Western blotting. Neither chlamydial development nor a secondary infection could be observed (*Figure 4—figure supplement 1H, I*). In addition, c-Myc was not stabilized upon chlamydial infection in the absence of Trp (*Figure 4—figure supplement 1H*), suggesting that viable bacteria and Trp are required for the stabilization of c-Myc. These data support the role of c-Myc in Trp- or indole-mediated rescue from IFN-γ-induced chlamydial persistence.

## Trp addition leads to activation of pGSK3β/c-Myc axis and restores chlamydial infection

Since we observed that Trp and c-Myc are both required for the development of *Chlamydia*, we next investigated the mechanism by which this amino acid regulates c-Myc levels. IFN-γ, upon binding to its receptor, activates phosphatidylinositol-3-kinase (PI3K) and serine-threonine protein kinase (AKT) and induces the dephosphorylation and activation of glycogen synthase kinase-3 (GSK3β) (*Nguyen et al., 2001*). If c-Myc is stabilized by the phosphorylation at serine 62 by the MAPK pathway, dephosphorylated active GSK3β phosphorylates c-Myc at threonine 58, followed by its ubiquitination and proteasomal degradation (*Albert et al., 1994*; *Figure 4D*). Both the MAPK and PI3K pathways activated during infection are critical for chlamydial development (*Capmany et al., 2019*; *Patel et al., 2014*; *Siegl et al., 2014*; *Subbarayal et al., 2015*). Therefore, the phosphorylation status of AKT and GSK3β was examined in HeLa 229 and human Fimb cells, which were treated with IFN-γ and/or Trp and infected with *Chlamydia*. IFN-γ activated the PI3 kinase pathway as evident from the phosphorylation of AKT (*Figure 4E*; *Figure 4—figure supplement 1J*). Despite this increase in AKT phosphorylation, IFN-γ treatment caused a reduction of GSK3β phosphorylation and c-Myc levels (*Figure 4E*; *Figure 4—figure supplement 1J*). Surprisingly, the addition of Trp to HeLa 229 or human Fimb cells increased phosphorylation of GSK3β, presumably resulting in its inactivation, and the elevation of c-Myc levels (*Figure 4E*; *Figure 4—figure supplement 1J*). Taken together, these data suggest that Trp rescues chlamydial infection via the activation of the pGSK3β/c-Myc axis.

## IFN-γ-induced downregulation of c-Myc has a pleiotropic effect on host metabolism

Since c-Myc is centrally involved in regulating amino acid transport (*Dong et al., 2020*), we asked if stabilized c-Myc also increases Trp uptake. In our previous RNA-seq analysis we observed the upregulation of the L-amino acid transporter Solute Carrier Family 7 Member 5 (LAT1/SLC7A5) in cells

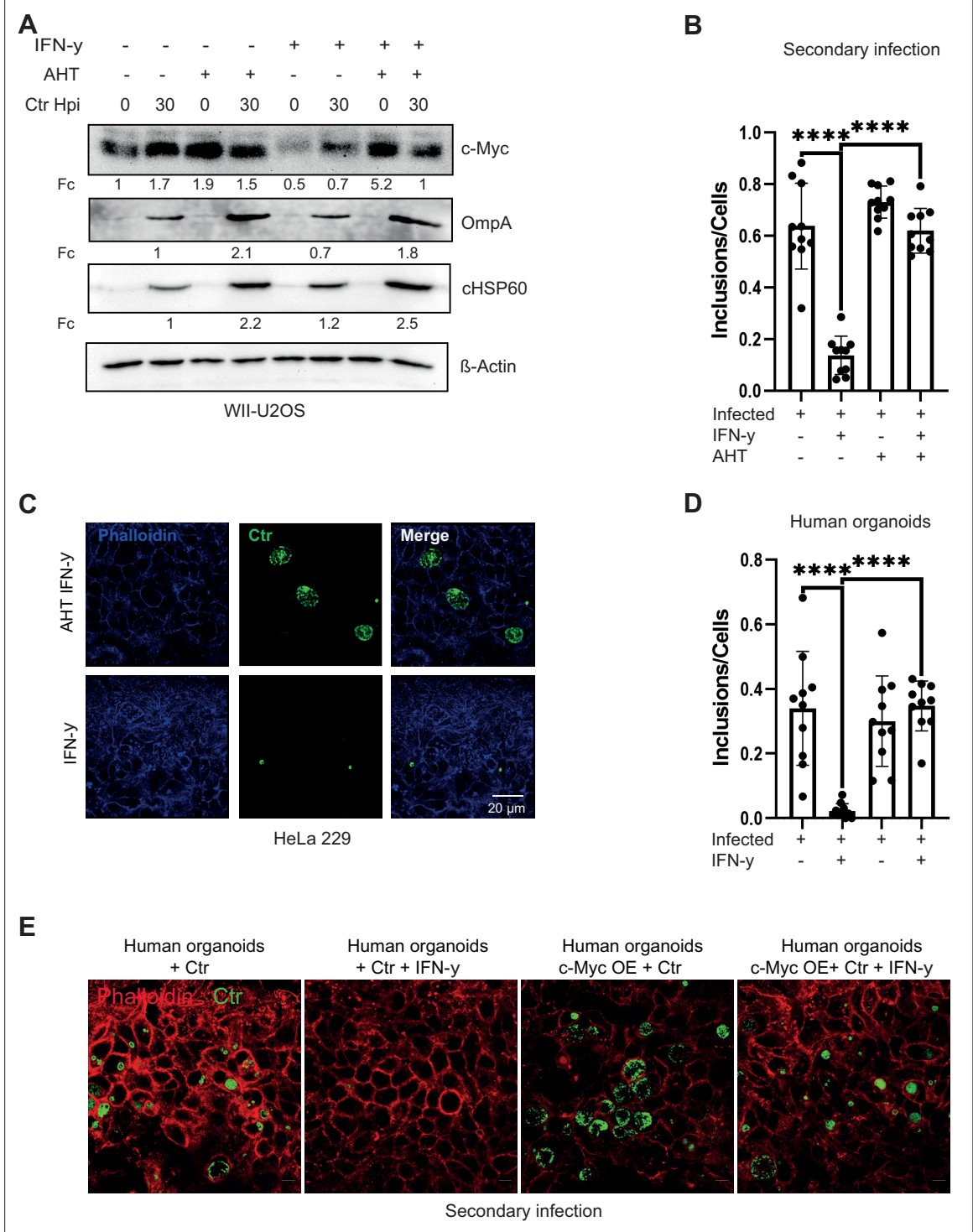

**Figure 3.** Expression of c-Myc rescues *Chlamydia* from persistence. (**A**) WII-U2OS cells were induced with 100 ng/mL anhydrotetracycline (AHT) for 2 hr. Cells were left untreated or were pre-treated for 2 hr with 10 ng/mL interferon-gamma (IFN-γ), infected with *Chlamydia* (*Ctr*) at multiplicity of infection (MOI) 1 and lysed at 30 hpi for Western blot analysis (n=3). (**B**) The infected cells from (**A**) were lysed to infect freshly plated HeLa 229 cells. The numbers of inclusions were counted from the different conditions shown in (**A**). The mean ± SD are shown in the graph. **** indicates a p<0.0001. (**C**) For the same culture conditions as in (**A**), an infectivity assay was performed. The HeLa 229 cells were fixed with 4% PFA after 30 hpi and immunostained for *Ctr* (cHSP60: green) and actin (Phalloidin: blue) in order to analyse the rescue. (**D**) From the experiment shown in (**E**) the infected/ IFN-γ-treated organoids were lysed with glass beads and different dilutions of the supernatant was used to infect freshly plated HeLa 229 cells. The number of inclusions were counted from three different experiments and mean ± SD are shown in the graph. **** indicates a p-value <0.0001. (**E**) The organoids from *Figure 3—*

*Figure 3 continued on next page*

*Figure 3 continued*

**figure supplement 1** were infected with *Ctr* for 6 days with and without IFN-γ, were lysed using glass beads, and dilutions of the supernatant were used to infect freshly plated HeLa 229 cells for an infectivity assay. Cells were fixed with 4% PFA after 30 hpi and immunostained for *Ctr* (cHSP60: green) and actin (Phalloidin: red) in order to analyse the rescue (n=3).

The online version of this article includes the following source data and figure supplement(s) for figure 3:

**Source data 1.** Complete and cutted membranes of all Western blots from *Figure 3*.

**Figure supplement 1.** Expression of c-Myc rescues *Chlamydia* from persistence.

**Figure supplement 1—source data 1.** Complete and cutted membranes of all Western blots from *Figure 3—figure supplement 1*.

infected with *Chlamydia* (**Rajeeve et al., 2020**). LAT1 is a system L-amino acid transporter with high affinity for branched chain and bulky amino acids, including Trp (**Bhutia et al., 2015**). Furthermore, the *LAT1* promoter has a binding site for c-Myc and it has been shown that overexpression of c-Myc leads to an increased expression of LAT1 (**Bhutia et al., 2015**). Therefore, the protein levels of LAT1 during a chlamydial infection and upon IFN-γ treatment were investigated by Western blotting. Accumulation of LAT1 protein was detected in a time-dependent manner in infected HeLa 229 cells (**Figure 5A**). In contrast, LAT1 protein levels were strongly reduced in IFN-γ-treated cells, irrespective of infection (**Figure 5B**). Following c-Myc expression, LAT1 levels in infected cells were rescued even in the presence of IFN-γ (**Figure 5C**). Differences in reduction of LAT1 levels between HeLa 229 and U2OS cells might be due to a better survival of c-Myc overexpression that can be pro-apoptotic in U2OS cells. Interestingly, the Trp-degrading enzyme IDO, which was strongly induced by IFN-γ treatment as expected, was only partially suppressed by c-Myc expression (**Figure 5C**). We then tested whether inhibition of IDO1 with the competitive inhibitor Epacadostat affects the levels of c-Myc. Whereas inhibiting IDO1 partially rescued *Ctr* from IFN-γ-induced persistence (**Figure 5D, E and F**), it had no effect on the levels of c-Myc (**Figure 5D**), indicating that rescue from IFN-γ-induced persistence by IDO1 inhibition does not depend on c-Myc levels.

To investigate how c-Myc restoration affects Trp metabolism in IFN-γ-treated cells, we measured intracellular levels of Trp via LC-MS of WII-U2OS cells with an AHT-inducible c-Myc expression. This analysis showed that *Chlamydia* infection increases intracellular levels of Trp, while addition of IFN-γ results in decreased Trp levels (**Figure 5G**). Interestingly, c-Myc expression in non-infected cells increased intracellular Trp to the levels of infected cells (**Figure 5G**). However, in the presence of IFN-γ, expression of c-Myc only induced a small increase in intracellular Trp levels to about the same level observed in uninfected cells (**Figure 5G**), indicating that c-Myc expression does not prevent the effect of IFN-γ on Trp degradation. To investigate whether induction of LAT1 by c-Myc increases Trp uptake, we also measured Trp levels in the culture medium (**Figure 5H**). Unexpectedly, c-Myc expression had no major effect on Trp uptake in untreated and IFN-γ-treated cells (**Figure 5H**). To directly measure the impact of c-Myc expression on the IDO activity, we determined the levels of kynurenine as the first and rate-limiting steps in Trp breakdown. To our surprise, c-Myc expression strongly enhanced the production of kynurenine in all IFN-γ-treated samples whereas infection alone or in cells with constant c-Myc expression had no effect (**Figure 5I**), suggesting that the relatively low Trp levels in the IFN-γ-treated, c-Myc-expressing cells depend on an increased turn-over. In conclusion, restoring Trp metabolism is not the main mechanism of the c-Myc-dependent rescue of chlamydial persistence downstream of IFN-γ signalling.

We next performed an unbiased metabolomics analysis. WII-U2OS cells were infected with *Chlamydia* and either induced with AHT and/or treated with IFN-γ, and the resulting changes in metabolite levels were analysed by LC-MS. Quality controls and data normalization were performed and a principal component analysis (PCA) demonstrated the validity of the datasets (**Figure 5—figure supplement 1A**). Interestingly, hierarchical clustering analysis revealed the grouping of all conditions permissive for chlamydial development (**Figure 6A**), strongly suggesting that these metabolite profiles are indicative of productive chlamydial infection. Chlamydial development is favoured in an environment with high levels of amino acids (tyrosine, histidine, alanine, homoserine, methionine, lysine, phenylalanine, threonine, asparagine, glutamate) and TCA (citrate, aconitate, malate, α-ketoglutarate) and urea cycle intermediates (ornithine, citrulline) (**Figure 6A**).

Pathway analysis revealed that nicotinate and nicotinamide metabolism are strongly regulated upon *Chlamydia* infection (**Figure 5—figure supplement 1B**). Moreover, phenylalanine, tyrosine,

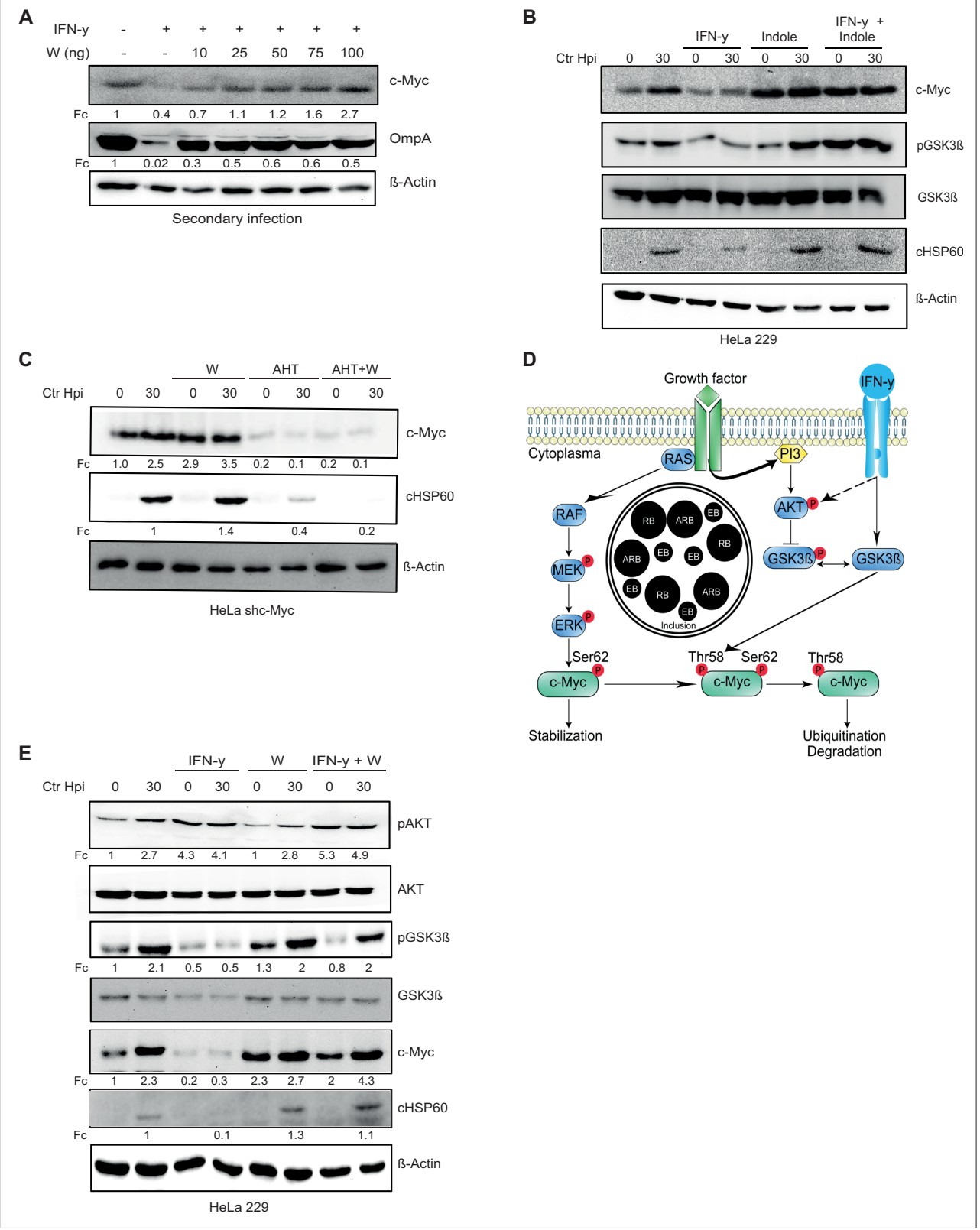

**Figure 4.** L-tryptophan activates the pGSK3β-c-Myc axis and rescues chlamydial infection. (**A**) HeLa 229 cells were treated with interferon-gamma (IFN-γ) and different doses of L-tryptophan (W) (10–100 ng/mL). After 30 hr the cells were lysed and used for Western blot analysis (n=3). (**B**) HeLa 229 cells were infected with *Ctr* at multiplicity of infection (MOI) 1 for 30 hr and treated with IFN-γ and/or indole. The cells were lysed and analysed by Western blotting for c-Myc, pGSK3β, and cHSP60 levels (n=3). (**C**) HeLa 229 cells with an anhydrotetracycline (AHT)-inducible expression of shc-Myc

*Figure 4 continued on next page*

*Figure 4 continued*

were either left uninfected or infected with *Chlamydia* at MOI 1 for 30 hr. Additionally, cells were treated with L-tryptophan and/or 100 ng/mL AHT to deplete c-Myc. The cells were further analysed via Western blotting (n=3). (**D**) Cartoon left side: active phosphatidylinositol-3-kinase (PI3K) with inactive glycogen synthase kinase-3 (GSK3β) leading to c-Myc stabilization. Right side: IFN-γ binding to its receptor, activates PI3K and serine-threonine protein kinase (AKT) and induces the dephosphorylation and activation of GSK3β, leading to c-Myc depletion. *Chlamydia* infection activates the PI3K and MEK/ERK pathway. (**E**) HeLa 229 cells were either left uninfected or infected with *Ctr* and treated with IFN-γ with or without L-tryptophan (W). The cells were analysed by Western blotting 30 hpi. cHSP60 shows the intensity of chlamydial infection and β-Actin serves as loading control (n=3). Western blots shown in **A-E** were quantified by normalizing the *Chlamydia* load (cHSP60 or OmpA) and the respective host cell protein levels to β-Actin and the result was indicated as fold change (Fc). Image of Western blots was taken from one of a total of at least three blots of biological replicates (n=3).

The online version of this article includes the following source data and figure supplement(s) for figure 4:

**Source data 1.** Complete and cutted membranes of all Western blots from *Figure 4*.

**Figure supplement 1.** L-tryptophan activates the pGSK3β-c-Myc axis and rescues chlamydial infection.

**Figure supplement 1—source data 1.** Complete and cutted membranes of all Western blots from *Figure 4—figure supplement 1*.

and tryptophan biosynthesis were among the significantly altered pathways with strongest impact upon infection, but also after IFN-γ treatment of infected and c-Myc overexpressing infected cells (*Figure 5—figure supplement 1B-D*). Other modulated metabolic pathways included amino acid pathways that have been shown before to be regulated upon *Chlamydia* infection cells (*Mehlitz et al., 2017*; *Rajeeve et al., 2020*). In addition, irrespective of IFN-γ treatment, arginine was one of the most depleted amino acids in infected cells (*Figure 6—figure supplement 1D*).

## α-Ketoglutarate and nucleosides rescue *Chlamydia* from IFN-γ-induced persistence

We next focused our attention on those metabolites that were induced by infection but reduced following IFN-γ treatment and restored by c-Myc overexpression as candidates that could be causally involved in *Chlamydia* persistence. *Chlamydia* infection resulted in a significant increase in several TCA cycle (related) intermediates, including aconitate, citrate, α-ketoglutarate, and glutamate (*Figure 6B*; *Figure 6—figure supplement 1A*). Furthermore, the amino acids aspartate, serine, and glycine, which function as important precursors for TCA cycle anaplerosis and nucleotide biosynthesis (*Figure 6—figure supplement 1C*), were also significantly induced by *Chlamydia* infection, while glutamine showed a trend towards induction that failed to reach statistical significance. In contrast, levels of arginine were strongly reduced upon infection, while intracellular levels of the urea cycle metabolites ornithine and citrulline were significantly increased (*Figure 6—figure supplement 1B, D*).

Remarkably, treatment of infected cells with IFN-γ lowered the induction of glutamate as well as aspartate, serine, and glycine. Moreover, levels of most TCA cycle metabolites, in particular aconitate, citrate, α-ketoglutarate, succinate, fumarate, and malate, were also reduced upon IFN-γ treatment, suggesting that reprogramming of host cell metabolism is a major part of the IFN-γ response (*Figure 6A and B*). Interestingly, re-expression of c-Myc restored levels of the TCA cycle metabolites citrate, aconitate and α-ketoglutarate, as well as the amino acids glutamine, glutamate, and glycine in IFN-γ-treated infected cells (*Figure 6A and B*; *Figure 6—figure supplement 1C*), suggesting that these metabolites are required for *Chlamydia* development.

We also investigated intracellular levels of purine and pyrimidine nucleotides and nucleosides as well as intermediates of nucleotide metabolism, as *Chlamydia* is an auxotroph for nucleotides. Interestingly, IFN-γ treatment significantly lowered the amounts of ATP and CTP as well as AMP, UMP, and cytidine (*Figure 6—figure supplement 1E, F*; *Figure 6C*). Moreover, several other metabolites involved in nucleotide metabolism showed a trend towards reduced abundance in IFN-γ-treated infected cells (*Figure 6—figure supplement 1E, F*; *Figure 6C*). This reduction in essential precursors for chlamydial DNA replication may explain why *Chlamydia* enter persistence in the presence of IFN-γ. In addition, overexpression of c-Myc enhanced levels of nucleoside triphosphates and strongly increased intracellular levels of adenosine, uridine, and cytidine in IFN-γ-treated cells (*Figure 6—figure supplement 1E*; *Figure 6C*). Another possibility is that IFN-γ-induced reduction of ATP levels generally slowed down metabolic processes and induced chlamydial persistence.

Based on the observation that IFN-γ leads to a reduction in TCA cycle intermediates and nucleotides in host cells, we investigated whether chlamydial growth after IFN-γ-induced persistence

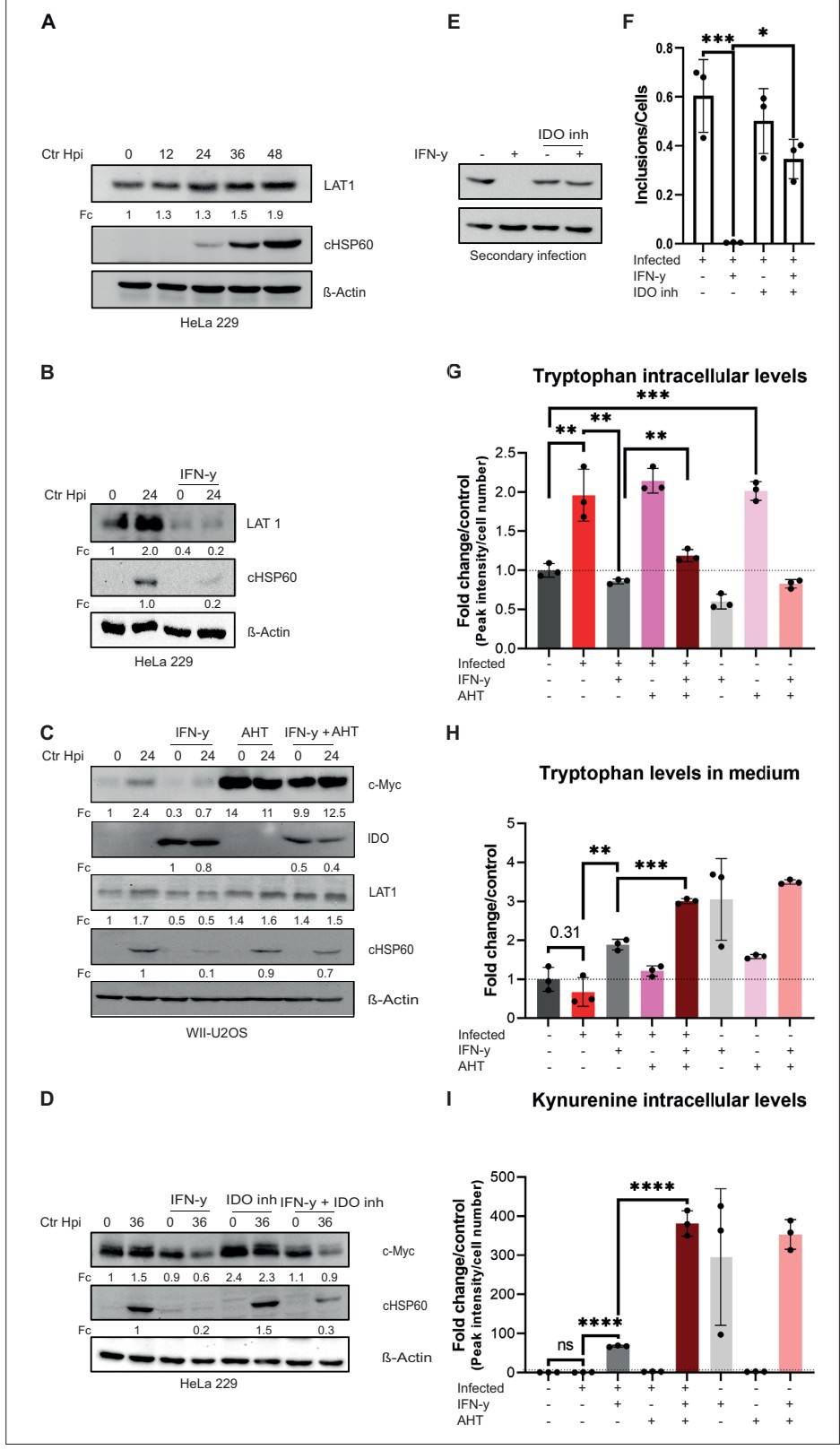

**Figure 5.** Influence of stabilized c-Myc on L-tryptophan (Trp) uptake and metabolites. (**A**) HeLa 229 cells were infected with *Chlamydia* (*Ctr*) for various time points. The cells were analysed via Western blotting for the levels of LAT1 (n=3). (**B**) HeLa 229 cells were either left untreated or were pre-treated for 2 hr with 10 ng/mL of IFN-γ, infected with *Ctr* at multiplicity of infection (MOI) 1 and lysed at 24 hpi to examine LAT1 regulation via Western

*Figure 5 continued on next page*

*Figure 5 continued*

blotting. (**C**) c-Myc overexpression in WII-U2OS cells was induced with 100 ng/mL anhydrotetracycline (AHT) for 2 hr. Cells were left untreated or were pre-treated for 2 hr with 10 ng/mL IFN-γ, infected with *Chlamydia* (*Ctr*) at MOI 1 and lysed at 24 hpi for Western blot analysis (n=3). (**D**) HeLa 229 cells were either left untreated or were pre-treated for 2 hr with 10 ng/mL of IFN-γ and/or 10 μM IDO inhibitor, infected with *Ctr* at MOI 1 and lysed at 36 hpi to examine c-Myc regulation via Western blotting. (**E**) After 48 hpi cells from (**D**) were lysed and used to infect freshly plated HeLa 229 cells to analyse progeny via Western blot. (**F**) The number of inclusions and the number of the cells of (**E**) were counted to plot the graph. One-way ANOVA was used for analysis. * indicates a p-value <0.05, *** indicates a p-value <0.001. (**G**) WII-U2OS cells were either left uninfected or infected with *Ctr* at MOI 1 for 30 hr. These cells were either left untreated or treated with just 10 ng/mL IFN-γ and/or 100 ng/mL AHT to induce expression of c-Myc. The cells were extracted, and metabolites were analysed by LC-MS. Data are presented as mean ± SD of triplicate wells. ** indicates a p-value <0.01, *** indicates a p-value <0.001. The intracellular levels of tryptophan are shown. (**H**) WII-U2OS cells were either left uninfected or infected with *Ctr* at MOI 1 for 30 hr. The infected cells were either left untreated or treated with just 10 ng/mL IFN-γ and 100 ng/mL AHT to induce overexpression of c-Myc. The media, in which the cells were grown, was extracted and metabolites were analysed by LC-MS. Data are presented as mean ± SD of triplicate wells. ** indicates a p-value <0.01, *** indicates a p-value <0.001. The levels of tryptophan present in medium are shown. (**I**) WII-U2OS cells were either left uninfected or infected with *Ctr* at MOI 1 for 30 hr. The infected cells were either left untreated or treated with just 10 ng/mL IFN-γ and 100 ng/mL AHT to induce expression of c-Myc. The cells were extracted, and metabolites were analysed by LC-MS. Data are presented as mean ± SD of triplicate wells. **** indicates a p-value <0.0001. The intracellular levels of kynurenine are shown. Western blots shown in Figure 5 were quantified by normalizing the *Chlamydia* load (cHSP60) and the respective host cell protein levels to β-Actin and the result was indicated as fold change (Fc). Image of western blots was taken from one of a total of at least three blots of biological replicates (n=3).

The online version of this article includes the following source data and figure supplement(s) for figure 5:

**Source data 1.** Complete and cutted membranes of all Western blots from *Figure 5*.

**Figure supplement 1.** Pathway analysis.

could be overcome by the supplementation of specific metabolic precursors. We therefore treated WII-U2OS and HeLa 229 cells with IFN-γ, followed by *Chlamydia* infection and addition of the cell-permeable dimethyl ester of α-ketoglutarate (DMKG) or a mixture of nucleosides (A, C, G, U). In both cell lines, the growth of *Chlamydia* was rescued by the addition of either DMKG or nucleosides and both conditions produced infectious progeny (*Figure 6D–F*). Moreover, restoration of chlamydial development could also be achieved by the sole addition of pyrimidine nucleosides (U+C), and to a minor extend also by purine nucleosides (A+G) (Western blot in *Figure 6F*). In contrast, purines seem to restore inclusion formation more efficiently than pyrimidines (bar diagram in *Figure 6F*). Furthermore, addition of citrate also showed the tendency to rescue chlamydial growth (*Figure 6G*). Even in primary human Fimb cells could the addition of DMKG restore the development of *Chlamydia* (*Figure 6H*). Taken together, these results indicate that IFN-γ broadly alters the metabolism of host cells to limit the availability of metabolic precursors for pyrimidine and purine biosynthesis to promote chlamydial persistence.

## The chlamydial Trp synthase pathway is not involved in c-Myc- and DMKG-mediated rescue of IFN-γ persistence

Since *Ctr* have an intact Trp synthase operon the effects of c-Myc on the rescue of IFN-γ-induced persistence could be indirect, for example, by the provision of precursors of the Trp synthase. We therefore generated a *trp*BA mutant in *Ctr* by Fluorescence-reported Allelic Exchange Mutagenesis (FRAEM) (*Keb and Fields, 2020*). This mutant lacked expression of TrpA or TrpB (*Figure 7A*) and was resistant to rescue from IFN-γ-induced persistence by generating Trp from indole (*Figure 7B*), in line with the phenotype of Trp synthase negative *Ctr* (*Caldwell et al., 2003*). However, expression of c-Myc restored growth of the *trp*BA mutant in the presence of IFN-γ (*Figure 7C*). In addition, the *trp*BA mutant also could be reactivated by the addition of exogenous DMKG to a similar extend as the wildtype strain (*Figure 7D*). These data show that the chlamydial Trp synthase is dispensable for the restoration of growth by c-Myc expression and supplementation of TCA metabolites in the presence of IFN-γ.

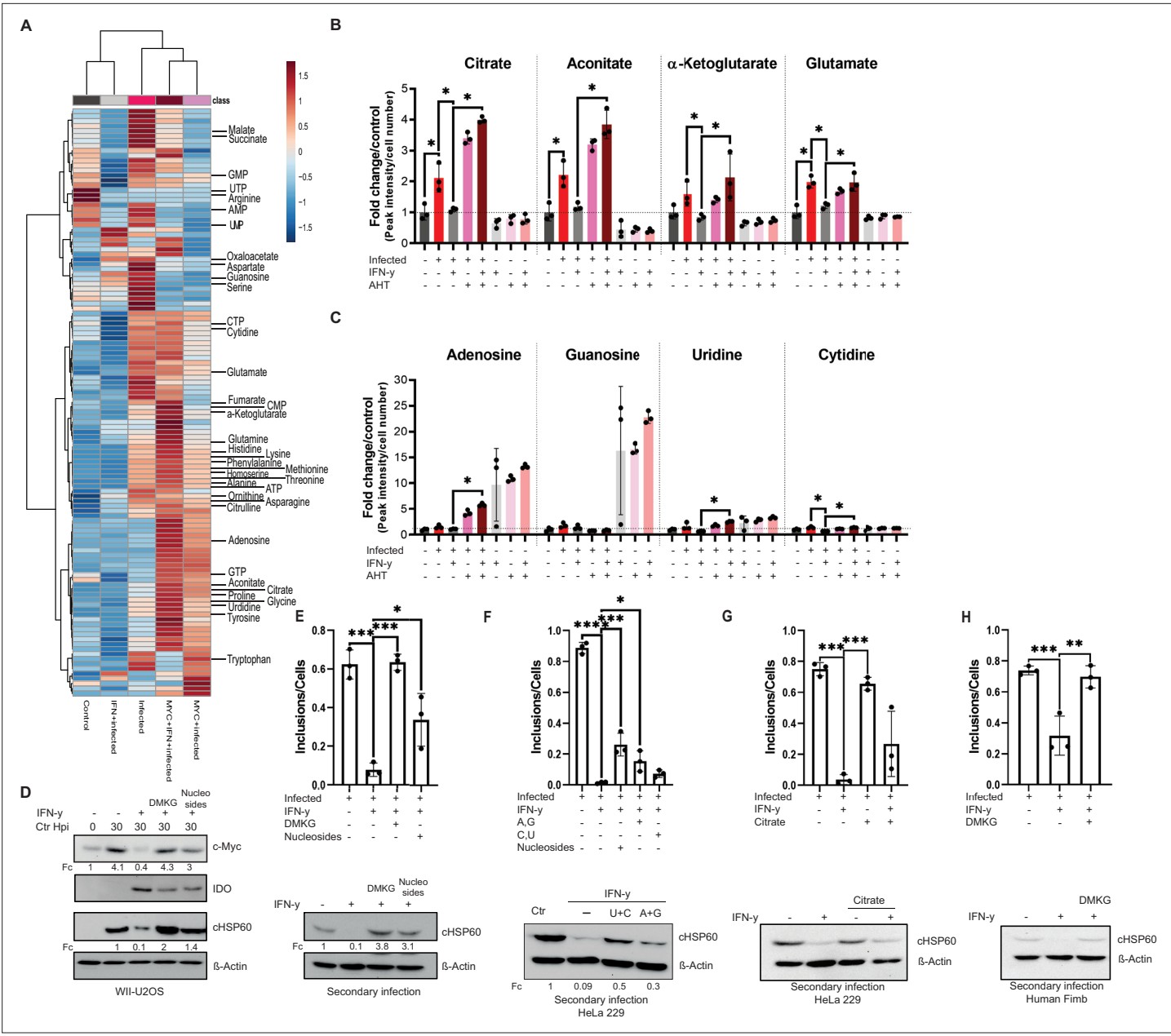

**Figure 6.** Tricarboxylic acid (TCA) intermediates and nucleosides can overcome interferon-gamma (IFN-γ)-induced persistence. (**A**) A heatmap with hierarchical clustering of all metabolites detected by LC-MS analysis of (*Figure 5G*). (**B/C**) WII-U2OS cells were either left uninfected or infected with *Chlamydia trachomatis (Ctr)* at multiplicity of infection (MOI) 1 for 30 hr. The infected cells were either left untreated or treated with just 10 ng/ mL IFN-γ and 100 ng/mL anhydrotetracycline (AHT) to induce overexpression of c-Myc. The cells were extracted, and metabolites were analysed by LC-MS. Data are presented as mean ± SD of triplicate wells. Intracellular levels of metabolites like citrate, aconitate, α-ketoglutarate, glutamate (**B**), or nucleosides (adenosine, guanosine, cytidine, and uridine) (**C**) were determined and quantified. Statistical analysis was performed using MetaboAnalyst 4.0. Significantly changed metabolite levels were determined by ANOVA with subsequent FDR correction, where a p value of 0.05 was considered statistically significant (*) and Tukey HSD was applied as post hoc test. (**D**) Cells were left untreated or were pre-treated for 2 hr with 10 ng/ mL IFN-γ, 4 mM α-ketoglutarate (DMKG), or 100 μM nucleosides, infected with *Ctr* at MOI 1 and lysed at 30 hpi for Western blot analysis (n=3). The α-ketoglutarate was supplied as cell-permeable dimethyl ester. (**E**) For the same culture conditions as in (**D**), an infectivity assay was performed. The cells were lysed at 48 hpi and dilutions of the resulting *Chlamydia* containing supernatant were added onto freshly plated WII-U2OS cells, which were then lysed at 30 hpi to examine infectious progeny via Western blotting (n=3). Inclusions and cells were counted, and the results are shown as bar diagram. One-way ANOVA was used for analysis. * indicates a p-value <0.05, *** indicates a p-value <0.001. (**F**) HeLa 229 cells were either infected with *Chlamydia* or infected and treated with IFN-γ and nucleosides (uridine/cytosine or adenosine/guanosine) were added. The cells were lysed at 48 hpi and dilutions of the resulting *Chlamydia* containing supernatant were added onto freshly plated HeLa 229 cells, which were then lysed at 30 hpi to examine infectious progenies via Western blot (n=3). Inclusions and cells were counted, and the results are shown as bar diagram. One-way ANOVA was used for

*Figure 6 continued on next page*

*Figure 6 continued*

analysis. * indicates a p-value <0.05, *** indicates a p-value <0.001, **** indicates a p-value <0.0001. (**G**) HeLa 229 cells were left untreated or were pre-treated for 2 hr with 10 ng/mL IFN-γ before 4 mM citrate and *Ctr* at an MOI 1 were added. The cells were lysed at 48 hpi and dilutions of the resulting *Ctr* containing supernatant were added onto freshly plated HeLa 229 cells, which were then lysed at 24 hpi to examine infectious progenies via Western blot (n=3). Inclusions and cells were counted, and the results are shown as bar diagram. One-way ANOVA was used for analysis. *** indicates a p-value <0.001. (**H**) Human Fimb cells were left untreated or were pre-treated for 2 hr with 10 ng/mL IFN-γ, 4 mM α-ketoglutarate (DMKG), infected with *Ctr* at MOI 1. The cells were lysed at 48 hpi and dilutions of the resulting *Chlamydia* containing supernatant were added onto freshly plated human Fimb cells, which were then lysed at 24 hpi to examine infectious progenies via Western blot (n=3). Inclusions and cells were counted, and the results are shown as bar diagram. One-way ANOVA was used for analysis. ** indicates a p-value <0.01, *** indicates a p-value <0.001. Western blots shown in Figure 6 were quantified by normalizing the *Chlamydia* load (cHSP60) and the respective host cell protein levels to β-Actin and the result was indicated as fold change (Fc). Image of Western blots was taken from one of a total of at least three blots of biological replicates (n=3).

The online version of this article includes the following source data and figure supplement(s) for figure 6:

**Source data 1.** Complete and cutted membranes of all Western blots from *Figure 6*.

**Figure supplement 1.** Urea cycle intermediates.

## Discussion

Persistent and recurrent infections are an important cause of excessive inflammation and tissue damage in the fallopian tube resulting in infertility and ectopic pregnancy (*Darville et al., 2003*; *Nagarajan et al., 2012*). Chlamydial infection of the epithelial cells lining the genital tract leads to the secretion of cytokines, like IL-8 (*Buchholz and Stephens, 2008*) and GM-CSF (*Lehr et al., 2018*), which attract myeloid and lymphoid cells towards the site of infection. This local immune response leads to chronic inflammation and may promote the development of malignancy including ovarian cancer (*Shan and Liu, 2009*). In addition, IFN-γ secreted by infiltrating T cells and NK cells provokes persistence of *Chlamydia* in epithelial cells. The current model of the immune defence elicited by IFN-γ in human cells centred around the induction of IDO and the consecutive degradation of Trp (*Beatty et al., 1994*; *Wyrick, 2010*). *Chlamydia* is auxotroph for Trp and enters persistence if this amino acid is degraded in the infected cell.

Our detailed molecular analysis of the IFN-γ-induced metabolic alterations in the host cells challenges this model and shifts the focus to the key transcription factor and proto-oncogene c-Myc as a central regulator of *Chlamydia* persistence. We provide evidence that IFN-γ acts via the GSK3β-STAT1 axis to deplete c-Myc levels and demonstrate that constitutive expression of c-Myc is sufficient to rescue *Chlamydia* from persistence induced by IFN-γ (*Figure 1*; *Figure 2*; *Figure 3*). Interestingly, this effect was also evident in human fallopian tube organoids, a newly developed physiologically relevant model for *Chlamydia* infection.

The pathway governing persistence uncovered in our study was indeed dependent on the levels of Trp (*Figure 4*; *Figure 4—figure supplement 1*), since *Chlamydia* failed to grow in the absence of this amino acid even in the presence of continuous c-Myc expression (*Figure 4—figure supplement 1H, I*). Conversely, the depletion of c-Myc and Trp addition was not sufficient to achieve normal chlamydial growth (*Figure 4C*; *Figure 4—figure supplement 1G*). Nevertheless, our results clearly show that both Trp and c-Myc are required for efficient bacterial replication (*Figure 4*; *Figure 4—figure supplement 1*). *Chlamydia* utilizes Trp to synthesize proteins, like the outer membrane protein MOMP, a Trp-rich polypeptide. Our findings that Trp but also indole, which *Ctr* can use as a substrate for Trp synthesis (*Byrne et al., 1986*; *Kari et al., 2011*; *Østergaard et al., 2016*), affects the level of c-Myc was unexpected (*Figure 4A and B*). It was shown previously that glutamine deprivation lowers levels of adenosine nucleotides and suppresses c-Myc in cancer cells by a mechanism dependent on the c-Myc 3'UTR (*Dejure et al., 2017*). We show here that Trp induces the phosphorylation and inactivation of GSK3β and thereby prevents the degradation of c-Myc by the ubiquitin proteasome system (*Figure 4*; *Figure 4—figure supplement 1*). Phosphorylation-dependent inactivation of GSK3β also occurs during normal chlamydial infection and causes the accumulation of c-Myc protein, as GSK3β leads to a destabilization of c-Myc by phosphorylation on the threonine 58 and proteasomal degradation (*AlZeer et al., 2017*; *Albert et al., 1994*). This finding suggests that Trp contributes to the rescue of *Ctr* from persistence by activating the pGSK3ß/c-Myc axis. Interestingly, c-Myc transcriptionally activates expression of the tryptophan transporter LAT-1, thereby increasing the uptake of Trp as part of a positive feedback regulation. Moreover, Trp depletion induced by IFN-γ signalling via STAT1 leads to loss of c-Myc expression, indicating that this amino acid functions as a central regulator of

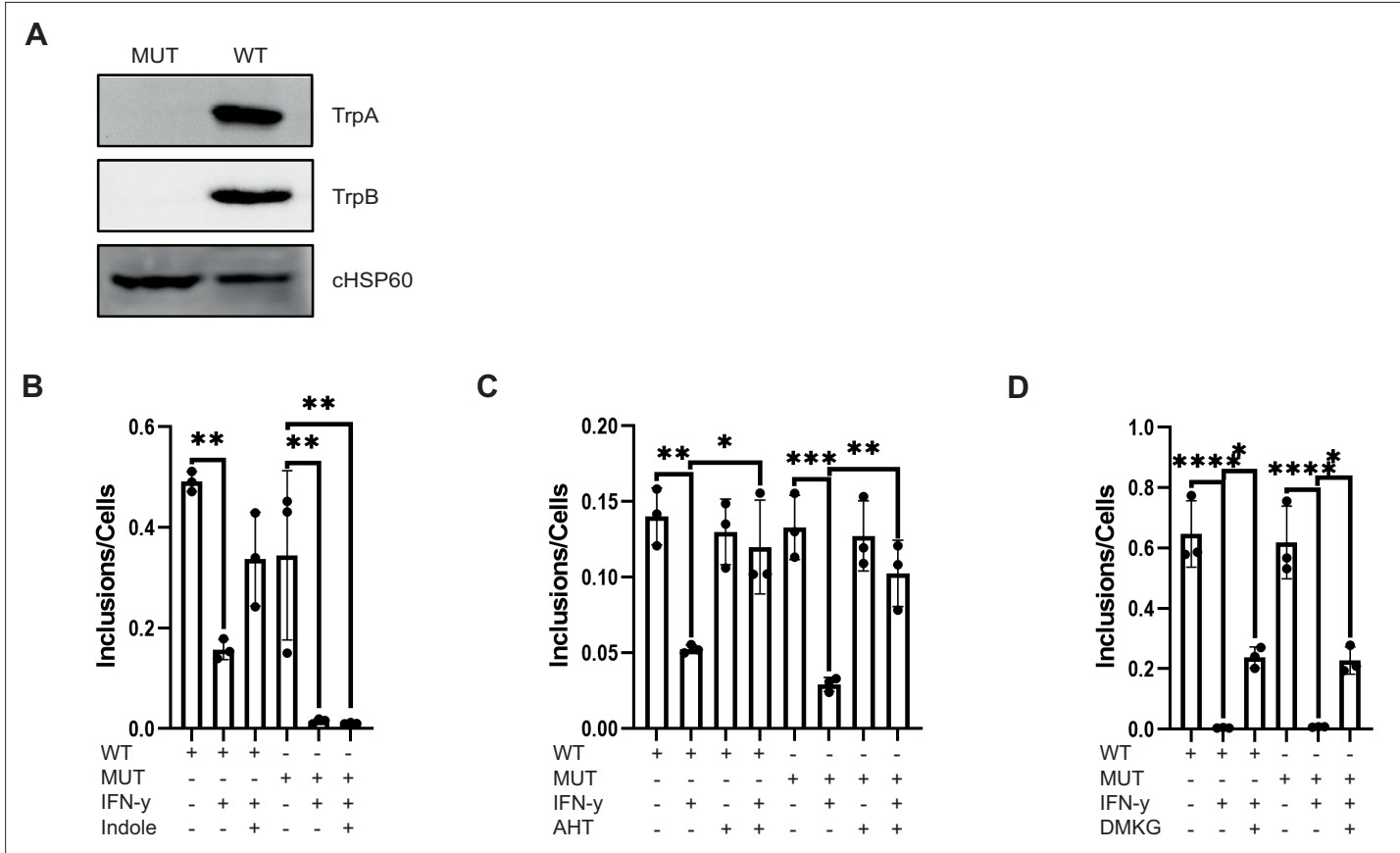

**Figure 7.** The chlamydial Trp synthase operon is not involved in the rescue from persistence by c-Myc expression and tricarboxylic acid (TCA) metabolite supply. (**A**) Western blot of wildtype *Chlamydia trachomatis (Ctr)* and TrpBA mutant to demonstrate the loss of TrpA and TrpB. (**B**) HeLa 229 cells were infected with *Ctr* or TrpBA mutant at multiplicity of infection (MOI) 1 for 48 hr and treated with 10 ng/mL IFN-γ and/or 50 µM indole. The cells were lysed, and the supernatant was used to infect freshly plated HeLa 229 cells to assess the infectivity of the progeny. The number of inclusions and the number of the cells were counted to plot the graph. One-way ANOVA was used for analysis. ** indicates p-value <0.01. (**C**) WII-U2OS cells were induced with 100 ng/mL anhydrotetracycline (AHT) for 2 hr. Cells were left untreated or were pre-treated for 2 hr with 10 ng/mL IFN-γ, infected with *Ctr* or TrpBA mutant at MOI 1 and lysed at 44 hpi for infecting freshly plated WII-U2OS cells to check infectious progeny. The number of inclusions and the number of the cells were counted to plot the graph. One-way ANOVA was used for analysis. * indicates a p-value <0.05, ** indicates p-value <0.01, *** indicates a p-value <0.001. (**D**) HeLa 229 cells were infected with *Ctr* or TrpBA mutant at MOI 1 for 48 hr and treated with 10 ng/mL IFN-γ and/or 4 mM α-ketoglutarate (DMKG). The cells were lysed, and the supernatant was used to infect freshly plated HeLa 229 cells to assess the infectivity of the progeny. Inclusions and cells were counted, and the results are shown as bar diagram. One-way ANOVA was used for analysis. * indicates p-value <0.05, **** indicates a p-value <0.0001.

The online version of this article includes the following source data for figure 7:

**Source data 1.** Complete and cutted membranes of all Western blots from *Figure 7*.

host cell metabolism and determines the decision between chlamydial development vs. the induction of persistence.

Detailed analysis of supernatants and extracts of infected cells also pointed to a broader impact of IFN-γ treatment on host cell metabolism. While Trp levels were significantly increased in infected cells and strongly reduced by IFN-γ treatment, constitutive expression of c-Myc failed to restore Trp levels in IFN-γ-treated cells despite preventing *Chlamydia* persistence (*Figure 5G and H*). Our data suggest that the highly efficient IDO prevents the restoration of Trp levels in the presence of IFN-γ even under constitutive c-Myc expression. Increased Trp levels induced by c-Myc expression are compensated by the enhanced degradation of Trp to kynurenine induced by IFN-γ (*Figure 5G and I*). All genital *Ctr* isolates possess Trp synthase activity which in the presence of indole mediates IFN-γ resistance (*Caldwell et al., 2003*). However, c-Myc expression also rescued a *trp*BA mutant from IFN-γ persistence (*Figure 7*) excluding a role of the Trp synthase and endogenous Trp synthesis in this pathway. This already indicated that additional metabolic pathways regulated by c-Myc expression

may be responsible for preventing persistence induced by IFN-γ. Unbiased metabolomics analysis of IFN-γ-treated, infected and c-Myc-expressing cells, and subsequent hierarchical clustering analyses revealed a grouping of all cells permissive for chlamydial replication (*Figure 6A*; infected, c-Myc infected, c-Myc IFN-γ infected), suggesting that this is the metabolic profile required for chlamydial replication.

Since c-Myc acts as a major metabolic regulator (*Dejure and Eilers, 2017*; *Stine et al., 2015*), we systematically investigated the c-Myc-dependent alterations in metabolite levels in response to bacterial infection and IFN-γ treatment. This analysis revealed that several intermediates of the TCA cycle, including citrate, aconitate, and α-ketoglutarate, as well as glutamate, were increased upon chlamydial infection, but significantly depleted upon IFN-γ treatment and restored when c-Myc was re-expressed (*Figure 6*). In addition, several nucleotides, in particular ATP and CTP, were among the top metabolites upregulated upon chlamydial infection (*Figure 6—figure supplement 1E*). Moreover, glutamate and alanine, which are both upregulated by infection, repressed in response to IFN-γ treatment and restored by c-Myc expression (*Figure 6A and B*), are central metabolites for *Chlamydia,* since they serve as precursors for cell wall biosynthesis (*Otten et al., 2018*). Interestingly, we could observe a strong increase in the levels of citrate in c-Myc overexpressing cells (*Figure 6B*). Citrate is required for the synthesis of fatty acids, which are scavenged by the bacteria from the host cell, as *Chlamydia* lack citrate synthase, aconitase, and isocitrate dehydrogenase and thus have only an incomplete TCA cycle (*Yao et al., 2014*; *Yao et al., 2015*). The finding of others that reduced uptake of glucose may play a role in IFN-γ-induced chlamydial persistence supports our model, since glycolysis fuels the TCA and nucleotide biosynthesis (*Shima et al., 2018*).

*Chlamydia*-infected cells showed a drastic reduction in the levels of arginine (*Figure 6—figure supplement 1D*). However, arginine levels were not restored upon c-Myc re-expression in IFN-γ-treated infected cells. It is possible that arginine is shuttled into the urea cycle, as we observed significantly higher levels of ornithine and citrulline (*Figure 6—figure supplement 1D*), which can be used for polyamine biosynthesis, that stabilize DNA and are essential for cell proliferation (*van Dam et al., 2002*). Further work is required to elucidate the exact role of arginine metabolism in chlamydial infection.

Here, we show a central role of c-Myc in the control of the host cell metabolism during IFN-γ-mediated innate immune defence against *Ctr* infection. The effect of IFN-γ-mediated c-Myc downregulation included substantial remodelling of host cell metabolism, including reduced abundance of TCA cycle intermediates, which are usually replenished via c-Myc-dependent glutaminolysis and anaplerosis. Thus, the enhanced production of amino acids, nucleotides and lipids, and, additionally, TCA cycle intermediates transferred from the host cell to the bacteria (*Mehlitz et al., 2017*; *Rajeeve et al., 2020*) may all be affected by IFN-γ treatment (*Kress et al., 2015*). The central role of host TCA cycle and nucleotide biosynthesis for chlamydial development is also supported by our finding that supplementing cell-permeable α-ketoglutarate or nucleosides can overcome the IFN-γ-induced persistent state of *Chlamydia* (*Figure 6*; *Figure 7D*). It can therefore be concluded that c-Myc is required for *Ctr* to induce metabolic reprogramming of the host cell to avoid nutrient shortage during replication and prevent the induction of persistence.

There is no doubt that Trp plays a central role in the control of chlamydial replication induced by IFN-γ (*Byrne et al., 1986*; *Taylor and Feng, 1991*). In the human host, depletion of Trp is the major effector mechanism of IFN-γ and the presence of a Trp synthase operon in the genital chlamydial isolates supports a central role of Trp in chlamydial persistence (*Fehlner-Gardiner et al., 2002*). However, our finding that both the substrate and the product of the Trp synthase, indole and Trp, control the levels of c-Myc suggests that Trp plays a role in signalling in chlamydial persistence beyond its sole function as an amino acid. IFN-γ depletes the essential amino acid Trp and causes a reduction in the levels of c-Myc, which affects the metabolic state of the host cell in a way that prevents chlamydial replication and induces persistence. It is very likely that similar mechanisms are also involved in IFN-γ-induced persistence of other intracellular bacteria (*Ganesan and Roy, 2019*). In addition, IFN-γ also restricts the infection of viruses (*Karupiah et al., 1993*; *Weizman et al., 2017*). Since c-Myc activation and metabolic reprogramming of cells is a prerequisite for the replication of several viruses (*Thai et al., 2014*; *Thai et al., 2015*), our findings may be generally relevant for the IFN-γ-dependent innate immune defence against infection.

## Materials and methods

### Cell lines and bacteria

Human Fimb cells (epithelial cells isolated from the fimbriae of patients undergoing hysterectomy), HeLa 229 cells (ATCC CCL-2.1) and HeLa 229 pInducer11 shc-Myc cells were cultured in RPMI1640+-GlutaMAX medium (Gibco) supplemented with 10% (v/v) heat-inactivated FBS (Sigma-Aldrich). WII-U2OS cells (*Lorenzin et al., 2016*) were maintained in high-glucose DMEM (Sigma-Aldrich) with 10% (v/v) heat-inactivated FBS. All cell lines were grown in a humidified atmosphere containing 5% (v/v) $CO_2$ at 37°C. In this study *Chlamydia trachomatis* serovar $L_2$/434/Bu (ATCC VR-902B), and *C. trachomatis* serovar D/UW-3/Cx (ATCC VR-885) were used and cultured and purified as published previously (*Paland et al., 2008*). In brief, *Chlamydia* were propagated in HeLa 229 cells at a multiplicity of infection (MOI) of 1 for 48 hr. Cells were mechanically detached and lysed using glass beads (3 mm, Roth). Low centrifugation supernatant (10 min at 2000× $g$ at 4°C) was transferred to high-speed centrifugation (30 min at 30,000× $g$ at 4°C) to pellet the bacteria. The pellet was washed and resuspended in ×1 SPG buffer (7.5% sucrose, 0.052% $KH_2PO_4$, 0.122% $Na_2HPO$, 0.072% L-glutamate). Aliquots were made, stored at –80°C, and the bacteria were titrated for MOI of 1 for use in further experiments. Infected cells were incubated in a humidified atmosphere with 5% (v/v) $CO_2$ at 35°C. The cell lines as well as the *Chlamydia* used in this study were tested to be free of *Mycoplasma* via PCR.

### Generation of *Ctr trp*BA mutant

The *C. trachomatis* serovar $L_2$/434/Bu *trp*BA mutants were generated by Fluorescence-reported Allelic Exchange Mutagenesis (FRAEM) as previously described (*Keb and Fields, 2020*; *Mueller et al., 2016*). Briefly, the up- and downstream fragments of *trp*A were amplified with primers trpA_us_fwd/trpA_us_rev (trpA up fwd: GCAGGTACCGGTCGACGAGAGCGGTTGAGTGCTATTTC; trpA up rev: AGTAGGAATGGTCGAAAATTCCTCTGTTTCTGCGGATG) and trpA_ds_fwd/trpA_ds_rev (trpA down fwd: TACGAAGTTATGACCTTTATGAATATGAATATGAAGCCCA; trpA down rev: CGGG GTCTGACGCCCGCTCTTGTTGGTTTCGGCAT) from *Ctr* genomic DNA. The upstream fragment was cloned into the unique *Sal*I site of the suicide vector pSUmC-4.0, whereas the downstream fragment was cloned into the unique SbfI restriction site of pSUmC-4.0. Thus, *trp*A up- and downstream fragments flanked the *aad*A and *gfp* genes in pSUmC-4.0 encoding for spectinomycin resistance and green fluorescent protein, respectively. The resulting vector was then transformed into *Ctr* using $CaCl_2$. McCoy cells were infected and transformants were selected by treatment with spectinomycin (*Keb and Fields, 2020*). Additionally, AHT selection was required for the expression of the plasmid replication factor pgp6. Bacteria were passaged every 48 hr onto fresh McCoy cells until GFP- and mCherry-positive clones appeared. These transformants also expressed mCherry encoded by the pSUmC-4.0 backbone. GFP- and mCherry-positive clones were selected and cultured in the absence of AHT. This resulted in plasmid loss and allowed selection for GFP-positive and mCherry-negative *trp*A mutants, where *trp*A was deleted through allelic exchange by the *aad*A-*gfp* cassette. The *trp*A mutant was FACS sorted for GFP. Deletion of *trp*A was verified by Western blot. Mutation of *trp*A also resulted in loss of *trp*B.

### Infectivity assay

For primary infection, control or IFN-γ-treated cells were infected with *C. trachomatis* at MOI 1 for 30–48 hr. The cells were lysed with glass beads and freshly plated cells were infected with different dilutions (referred to as secondary infection). Lysates of the secondary infection were taken to determine the infectivity via Western blotting against chlamydial HSP60.

### Induction and chemicals

To induce c-Myc overexpression in WII-U2OS cells or c-Myc knockdown in HeLa 229 pInducer11 shc-Myc cells, medium was changed to 10% (v/v) heat-inactivated FBS DMEM or RPMI accordingly and 100 ng/mL AHT (Acros) was added. After 2 hr of induction, cells were infected with *C. trachomatis* and incubated for further 24–36 hr. Likewise, IFN-γ (Gibco, Merck Millipore), interferon-alpha (Merck Millipore), Trp (Sigma-Aldrich), Indole (Sigma-Aldrich), citrate (Roth), and IDO inhibitor Epacadostat (Biozol) were added to cells as needed 2 hr before infection.

## Induction of persistence

Cells and organoids were pre-treated with IFN-γ (Recombinant Human IFN-γ PHC4031 Gibco, IF002 Merck Millipore) for 2 hr at least prior infection (*Nelson et al., 2005*). Throughout the experiments, according amounts of IFN-γ were left in the medium. HeLa 229 cells were pre-treated for 2 hr with 1 unit of penicillin (Penicillin G sodium salt 13752-5G-F Sigma-Aldrich) before infection and left in the medium during the whole infection period.

## Western blot and antibodies

For Western blot analysis, cells were directly lysed with ×2 Laemmli buffer (10% 1.5 M Tris-HCl pH 6.8, 4% SDS, 30% glycerol, and 1.5% β-mercaptoethanol) on ice. Protein samples were separated with a 10% SDS-PAGE (Peqlab) and transferred onto a PVDF membrane (Sigma-Aldrich) via a semi-dry blotter (Peqlab) (2 hr at 1 mA/cm$^2$). The membrane was blocked in 5% (w/v) non-fat dried milk powder in ×1 Tris-buffered saline with 0.5% Tween20 (Sigma-Aldrich) for 1 hr and then incubated in the appropriate primary antibody overnight at 4°C. Antibodies against c-Myc, c-Myc phospho-T58, and c-Myc phospho-S62 were purchased from Abcam. Phospho-Akt (Ser473), Akt (pan), phospho-Erk, Erk, phospho-GSK-3β (Ser9), GSK-3β (3D10), phospho-Stat1 (Ser727), phospho-Stat1 (Tyr701), and Stat1 antibodies were obtained from Cell Signaling. β-Actin antibody was purchased from Sigma-Aldrich. Chlamydial HSP60 antibody was obtained from Santa Cruz. An IDO antibody was kindly provided by Ida Rosenkrands and OmpA was self-made. Proteins were detected with corresponding horseradish peroxidase-conjugated secondary antibody (Santa Cruz), using homemade ECL solutions and Intas Chemiluminescence Imager.

## Immunofluorescence analysis

The cells were seeded on cover slips and infected with *C. trachomatis* serovar L$_2$ at MOI 1 for indicated time points. Before fixation with 4% PFA/Sucrose (Roth), the cells were washed with DPBS (Gibco). Fixed cells were permeabilized with 0.2% Triton X-100 (Sigma-Aldrich) in ×1 DPBS for 30 min, blocked with 2% FBS in ×1 DPBS for 45 min and incubated with primary antibodies for 1 hr at room temperature. Primary antibodies, chlamydial HSP60 (Santa Cruz, 1:500) and Phalloidin (Thermo Fisher Scientific), were diluted in 2% FBS in ×1 DPBS. Samples were washed and incubated with fluorescence dye conjugated secondary antibodies (Dianova) for 1 hr in the dark at room temperature. Cover slips were mounted onto microscopy slides using mowiol, slides were air-dried for at least 24 hr and examined using a LEICA DM2500 fluorescence microscope. The images were analysed with LAS AF program (Leica) and ImageJ software.

## Metabolic profiling

For this study, 10$^6$ WII-U2OS cells per well (six-well culture plate, Costar) were seeded in triplicates, either uninfected or infected with *C. trachomatis* serovar L$_2$ for 30 hr. c-Myc overexpression was induced by AHT and cells were treated with IFN-γ as described above. After the respective time, medium was collected, snap-frozen in liquid nitrogen, and the cells were washed with ice-cold 154 mM ammonium acetate (Sigma) and snap-frozen in liquid nitrogen. The cells were harvested after adding 480 µL cold MeOH/H$_2$O (80/20, v/v) (Merck) to each sample containing Lamivudine (Sigma) standard (10 µM). The cell suspension was collected by centrifugation and transferred to an activated (by elution of 1 mL CH$_3$CN [Merck]) and equilibrated (by elution of 1 mL MeOH/H$_2$O [80/20, v/v]) C18-E SPE-column (Phenomenex). The eluate was collected and evaporated in SpeedVac and was dissolved in 50 µL CH$_3$CN/5 mM NH$_4$OAc (25/75). Each sample was diluted 1:1 (cells) or 1:5 (medium) in CH$_3$CN. Five µL of sample was applied to HILIC column (Acclaim Mixed-Mode HILIC- 1, 3 µm, 2.1 * 150 mm). Metabolites were separated at 30°C by LC using a DIONEX Ultimate 3000 UPLC system (Solvent A: 5 mM NH$_4$OAc in CH$_3$CN/H$_2$O (5/95), Solvent B: 5 mM NH$_4$OAc in CH$_3$CN/H$_2$O (95/5); Gradient: linear from 100% B to 50% B in 6 min, followed by 15 min const. 40% B, then returning to 100% B within 1 min) at a flow rate of 350 µL/min. After chromatographic separation, masses (m/z) were acquired using a Q-Exactive instrument (Thermo Fisher Scientific) in positive and negative ionization mode in the scan range between 69 and 1000 m/z with a resolution of 70,000. AGC target was set to 3×10$^6$ and maximum injection time was set to 200 ms. Sheath, auxiliary, and sweep gas were set to 30, 10, and 3, respectively, and spray voltage was fixed to 3.6 kV. S-lens RF level was set to 55.0, the capillary and the Aux gas heater were heated to 320°C and 120°C, respectively. Peak determination

and semi-quantitation were performed using TraceFinder software. Obtained signal intensities were normalized to the internal standard (lamivudine) and a cell number, which was determined by crystal violet staining. In brief, for the crystal violet staining cells were fixed with 4% PFA/Sucrose (Roth), stained with 0.1% crystal violet (Merck) dissolved in 20% ethanol (Roth), washed with water, dried overnight, and the absorbance was measured at 550 nm. The pellet of the cell samples was dried, resuspended in 0.2 M sodium hydroxide (Roth), cooked for 20 min at 95°C, and absorbance was measured at 550 nm. Statistical analysis was performed using GraphPad Prism or MetaboAnalyst 4.0. Significantly changed metabolite levels were determined by ANOVA with subsequent FDR correction, where a p value of 0.05 was considered statistically significant (*) and Tukey HSD was applied as post hoc test. Hierarchical clustering, PCA, and pathway analyses were done and resulting plots were generated by MetaboAnalyst 4.0 (*Chong et al., 2019*).

### Generation and culture of human fallopian tube organoids

Generation of human organoids was adapted from *Kessler et al., 2019*. Fallopian tube tissue was obtained from patients undergoing hysterectomy. The tissue was processed within 2 hr after the surgery. Briefly, tissue samples were washed with DPBS (Gibco), placed into a sterile Petri dish (Corning), cut into small pieces, and pressed against the dish with a glass slide (VWR). The cells were washed with DPBS, incubated with 1 mg/mL collagenase type I (Sigma) for one hour at 37°C to dissociate epithelial cell, centrifuged at 1000× $g$ for 10 min. The supernatant was removed, the pellet was resuspended in Matrigel (Corning) and plated in 50 µL drops in a 24-well plate. After 20 min of incubation at 37°C, organoid growth medium (Advanced DMEM [Thermo Fisher Scientific], 25% conditioned Wnt3A medium, 25% conditioned RSPO1 medium, 10% conditioned Noggin medium, 2% B27 [Thermo Fisher Scientific], 1% N$_2$ [Thermo Fisher Scientific], 1 mM nicotinamide [Sigma], 50 ng/mL human EGF [Thermo Fisher Scientific], 100 ng/mL human FGF [Thermo Fisher Scientific], 0.5 mM TGF-β R Kinase Inhibitor IV [Tocris], 10 mM ROCK inhibitor [Abmole Bioscience]) was added to the wells.

Organoids were splitted every 1–2 weeks, at a ratio of 1:2. The Matrigel drop was resuspended in cold Advanced DMEM medium, mechanically fragmented with a 26 G needle, and centrifuged at 1000× $g$ at 4°C for 5 min. The pellet was resuspended in the respective amount of fresh Matrigel and seeded in 50 µL drops in cell culture well plates added and further processed as explained above.

### Infectivity assay in organoids

For infection with *Ctr* mature organoids were released from Matrigel drops by resuspending them in ice-cold DPBS (Gibco), pooled, mechanically fragmented with a needle, and distributed in equal amounts in Eppendorf tubes. The IFN-γ-treated or untreated organoids were infected with *Ctr* L$_2$ (5×10$^5$ IFU). Afterwards Matrigel was added to the pellets and organoids were seeded in a cell culture plate. Six days post infection the organoids were fixed with 4% PFA and used for immunostaining. In addition, infected organoids were lysed with glass beads and different dilutions were used to infect freshly plated HeLa 229 cells to analyse the infectivity of the progeny.

## Acknowledgements

We thank Dr Jörg Wischhusen for providing the human Fimb cells, Dr Francesca Dejure for the WII-U2OS cells, and Dr Ken Fields for pSUmC-4.0 and pSUmC-CRE-bla plasmids. We thank Dr Adriana Moldovan and Marcel Rühling for technical assistance. We thank Dr Andreas Demuth for critically reading the manuscript. This work was supported by the Deutsche Forschungsgemeinschaft (DFG) priority program GRK2157 '3D Tissue Models for Studying Microbial Infections by Human Pathogens' and the European Research Council (grant no. ERC-2018-ADG/NCI-CAD) to TR.

## Additional information

### Funding

| Funder | Grant reference number | Author |
|---|---|---|
| European Research Council | ERC-2018-ADG/NCI-CAD | Thomas Rudel |
| Deutsche Forschungsgemeinschaft | RTG 2157/2 | Thomas Rudel |

The funders had no role in study design, data collection and interpretation, or the decision to submit the work for publication.

### Author contributions

Nadine Vollmuth, Conceptualization, Formal analysis, Investigation, Writing – original draft; Lisa Schlicker, Formal analysis, Methodology, Writing – review and editing; Yongxia Guo, Pargev Hovhannisyan, Naziia Kurmasheva, Investigation; Sudha Janaki-Raman, Formal analysis, Investigation, Writing – review and editing; Werner Schmitz, Formal analysis; Almut Schulze, Formal analysis, Writing – review and editing; Kathrin Stelzner, Supervision; Karthika Rajeeve, Conceptualization, Supervision, Investigation, Writing – original draft; Thomas Rudel, Conceptualization, Formal analysis, Supervision, Funding acquisition, Writing – original draft, Project administration, Writing – review and editing

### Author ORCIDs

Nadine Vollmuth http://orcid.org/0000-0002-1003-4410
Lisa Schlicker http://orcid.org/0000-0002-8350-2289
Thomas Rudel http://orcid.org/0000-0003-4740-6991

### Decision letter and Author response

Decision letter https://doi.org/10.7554/eLife.76721.sa1
Author response https://doi.org/10.7554/eLife.76721.sa2

## Additional files

### Supplementary files

• Transparent reporting form
• Source data 1. Inclusion Forming Units (IFU).
• Source data 2. Metabolic profiling.

### Data availability

Metabolomics data set has been uploaded with the manuscript.

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
