## [Editor Report]

This paper will be of interest to scientists working to understand *Chlamydia trachomatis* persistence, and host pathogen interaction in general. The authors report the surprising observation that the mechanism of restriction of bacterial growth is through the inhibition of c-Myc signaling by IFNγ as opposed to IDO-dependent depletion of tryptophan levels, as had been previously suggested.

---

## [Decision Letter]

**Decision letter after peer review:**

Thank you for submitting your article "Myc plays a key role in IFN-γ-induced persistence of *Chlamydia trachomatis*" for consideration by *eLife*. Your article has been reviewed by 3 peer reviewers, and the evaluation has been overseen by a Reviewing Editor and Dominique Soldati-Favre as the Senior Editor. The reviewers have opted to remain anonymous.

Essential revisions:

Data need strengthening in terms of (a) quantification, (b) statistical analyses, (c) controls and (d) consistency of models used.

(a) Quantification:

Strengthen all instances of quantification based on western blots: a single Western blot is not acceptable evidence. Report the level of variance for technical replicates, or variability among independent biological replicates. (See for example Figure 4B, E; S4C, H; Figure S1C.)

In Figure 1B, there is a significant increase in c-Myc protein in condition 2 compared to condition 1 (~300%). However, in Figure S1C, the increase from t0 to 30 hpi, appears to be very mild (~50%) in c-Myc protein levels compared with at 0 hpi (in the non-IFNγ-treated samples). In light of these variations, please report on the reproducibility of Ctr-mediated c-Myc induction.

Figures 4A-C, S4A-B, S4D. The Western blots showing cHSP60/OmpA levels are not alone sufficiently convincing to support that tryptophan or indole supplementation rescues chlamydial growth after IFNγ treatment. Please provide another method to support these statements (e.g. microscopy).

(b) Statistical analyses:

Provide a systematic description of the statistical tests used throughout.

More specifically, do results of Figure 2D hold when an unpaired t-test rather than a paired t-test is used for the analysis? Lines 152-156. Statistical analyses should be carried out between untreated and IFNγ/AHT-treated samples in Figures S1I and S1J respectively to support the statements described here. Provide error bars, number of repeats and statistical analysis for figure S3B. Figures 6B-C, S6C-F, clarify the choice of statistical tests. Should multiple testing correction be applied considering the nature of these experiments/analyses?

(c) Controls:

Figure 1A-C, please report if, at 24 hpi for conditions 3 and 4, Ctr inclusion formation and c-Myc levels were, as expected, the same. In addition, provide a control wherein cells expressing a control shRNA that AHT expression does not impact Ctr inclusion formation in the absence of c-Myc suppression.

Figure S4F-G. Include a control in which cHSP60 is expected/can be detected.

The addition of DMKG can rescue Ctr from IFN-γ-induced persistence. Please test the addition of aconitate and citrate, both metabolites that should be required for *C. trachomatis* development too.

Type I IFNS do not induce persistence and only weakly limit Chlamydia replication, yet they also signal through STAT1. Test the model for STAT1-dependent regulation of c-myc. If IFNa/b decreases c-myc levels yet causes no persistence, reconsider the model linking IFNγ-induced tryptophan limitation to destabilisation of c-Myc.

(d) Consistency:

There are differences in response to IFN γ between cell lines that are not addressed fully.

Figure S3 shows c-Myc overexpression in HeLa cells but not in organoids, provide this result.

Similarly, the degree to which DMKG and nucleosides affect persistence appears less consistent (in contrast to L-tryptophane) and possibly confounded by different cell lines. In Figure 6, the addition of DMKG to IFN-treated cells during infection rescues inclusion number to levels of the untreated control but significantly less in Figure 7D. The addition of nucleosides in the cells used in Figure 6 rescues persistence significantly less (as assessed by inclusion number) than DMKG, but there is no data presented from the cells used in Figure 7. The organoid infection experiments seemed to be underpowered and quantification of infectivity is a little suspect (Figure S2H- inclusions were titers an uneven number of cells). U2OS and HeLa cells are transformed and their metabolism is likely to be very distinct. Thus, show if DMKG or nucleosides rescue persistence in primary Fimb cells.

In addition, the reduction of LAT1 levels is quite striking in HeLa cells (Figure 5C) but very minor in U2OS cells. U2OS cells might survive better c-Myc overexpression that can be pro-apoptotic. Please acknowledge.

(e) In addition, the model linking IFNγ-induced tryptophan limitation to destabilisation of c-Myc via PI3K-GSK3b is not fully supported by the current evidence. Either provide direct evidence of either increased phosphorylation of threonine 58 on c-Myc or increased proteasomal degradation of c-Myc as a result of decreased levels of tryptophan upon IFNγ treatment (at the moment, it seems that phosphorylation of threonine 58 is unchanged upon IFNγ treatment (see Lines 157-162 and Figure 1E)); or remove the model and rework the text accordingly. Of note, bacterial replication also induces AKT and GSK3b.

(f) Line 541-544/Figure S6E: Considering that the most pronounced effect appears to be on adenosine (Figure 6C) and ATP (Figure S6E), please consider and discuss that the effect could be mediated by reduced ATP levels generally slowing down metabolic processes rather than DNA replication in particular.

(g) Line 553-555/Figure 6F. The authors suggest that "restoration of chlamydial development could also be achieved by the sole addition of pyrimidine (U+C), and to a minor extend [sic] also purine nucleosides (A+G)". However, Figure 6F appear to suggest that purines (A+G) have a more significant contribution to *Chlamydia trachomatis* development than pyrimidines (C+U), which appear to have a statistically insignificant effect. Please clarify.

---

## [Author Response]

Essential revisions:Data need strengthening in terms of (a) quantification, (b) statistical analyses, (c) controls and (d) consistency of models used.(a) Quantification:Strengthen all instances of quantification based on western blots: a single Western blot is not acceptable evidence. Report the level of variance for technical replicates, or variability among independent biological replicates. (See for example Figure 4B, E; S4C, H; Figure S1C.)

We fully agree that a single western blot is not sufficient to draw any conclusions. We generally repeat western blots three times from three biological replicates. We now explicitly mention this in the figure legend of figures showing western blots. Generally, the tendency of the results shown in the manuscript was always the same, individual levels of protein bands varied in individual blots probably due to blotting efficiency.

In Figure 1B, there is a significant increase in c-Myc protein in condition 2 compared to condition 1 (~300%). However, in Figure S1C, the increase from t0 to 30 hpi, appears to be very mild (~50%) in c-Myc protein levels compared with at 0 hpi (in the non-IFNγ-treated samples). In light of these variations, please report on the reproducibility of Ctr-mediated c-Myc induction.

Variation of c-Myc induction dependent on the infection conditions and the cell line or cell type used is indeed what we see. In the case mentioned here, the cell lines are different. In Figure 1B, HeLa229-shc-Myc, in the experiment shown in Figure S1C (new Figure 1 —figure supplement 1E), the cell was HeLa229 which responds differently to *Chlamydia* infection.

Figures 4A-C, S4A-B, S4D. The Western blots showing cHSP60/OmpA levels are not alone sufficiently convincing to support that tryptophan or indole supplementation rescues chlamydial growth after IFNγ treatment. Please provide another method to support these statements (e.g. microscopy).

We included light microscopy pictures to demonstrate the clear difference in inclusion formation dependent on tryptophan (Figure 4 —figure supplement 1C) and indole (Figure 4 —figure supplement 1F) supplementation after IFN-γ treatment. Unfortunately, the quality of the images is low in the pdf due to the size restrictions of the journal for the upload of manuscripts. We are happy to provide high resolution images if requested by the reviewers.

(b) Statistical analyses:Provide a systematic description of the statistical tests used throughout.More specifically, do results of Figure 2D hold when an unpaired t-test rather than a paired t-test is used for the analysis?

We provided a detailed description of the statistical test in the manuscript in each figure legend. If we use an unpaired t-test for the data shown in Figure 2D, the differences are not significant (0.1268; 0.6641). However, we used the paired t test since the organoids are all derived from the same biopsy. For this experimental setting, the paired t-test should be appropriate.

Lines 152-156. Statistical analyses should be carried out between untreated and IFNγ/AHT-treated samples in Figures S1I and S1J respectively to support the statements described here.

We performed several statistical analyses (unpaired t-test, paired t-test and one-way ANOVA) and none of them revealed significant induction of *trp*B. (Figure S1I and S1J new Figure 1 —figure supplement 1K and L)

Provide error bars, number of repeats and statistical analysis for figure S3B.

We replaced Figure 3 —figure supplement 1B and included error bars and the number of individual repeats.

Figures 6B-C, S6C-F, clarify the choice of statistical tests. Should multiple testing correction be applied considering the nature of these experiments/analyses?

We are grateful to the reviewers for this comment and fully agree that multiple testing correction should be applied for these data. We repeated the analysis and included FDR correction. Figure 6B-C and Figure 6 —figure supplement 1C-F have been exchanged.

In the methods part we included analysis as follows: “Statistical analysis was performed using Prism GraphPad or MetaboAnalyst 4.0. Significantly changed metabolite levels were determined by ANOVA with subsequent FDR correction, where a p value of 0.05 was considered statistically significant and Tukey HSD was applied as post-hoc test.”

(c) Controls:Figure 1A-C, please report if, at 24 hpi for conditions 3 and 4, Ctr inclusion formation and c-Myc levels were, as expected, the same. In addition, provide a control wherein cells expressing a control shRNA that AHT expression does not impact Ctr inclusion formation in the absence of c-Myc suppression.

At 24 hpi, AHT was remove and the cells were washed with cell culture medium. At this time point, the inclusions and c-Myc levels were the same. This is expected since the samples were treated identical to this point. We have added this statement to the text on page 6.

For the control shRNA experiment we included HeLa229 NTC SCC5 cells which express a control shRNA (pInducer11 non-targeting shRNA). Treatment of this cell line under the same conditions as the knockdown cell line revealed no difference. IF images and IFU counts of these experiments were added in Figure S1A and S1B, respectively.

Figure S4F-G. Include a control in which cHSP60 is expected/can be detected.

We included a control in which chlamydial replication is possible (addition of tryptophan) and cHsp60 expression can be detected. These blots are now included in the new Figure 4 —figure supplement 1H and I.

The addition of DMKG can rescue Ctr from IFN-γ-induced persistence. Please test the addition of aconitate and citrate, both metabolites that should be required for *C. trachomatis* development too.

We performed the experiment with citrate as suggested and detected the trend to rescue as predicted by the reviewer. The results have been included as a new Figure 6G showing IFU data and a western blot. We unfortunately could not perform the aconitate experiment since the order was postponed by the company. The expected delivery is now in August 2022. Since the delivery in August is not guaranteed and we were already delayed with the revision of the manuscript, we decided to submit the revision without the aconitate data.

Type I IFNS do not induce persistence and only weakly limit Chlamydia replication, yet they also signal through STAT1. Test the model for STAT1-dependent regulation of c-myc. If IFNa/b decreases c-myc levels yet causes no persistence, reconsider the model linking IFNγ-induced tryptophan limitation to destabilisation of c-Myc.

We performed the experiment with IFN-α as suggested by the reviewers and tested the effects on c-Myc and *Ctr.* C-Myc level were slightly reduced compared to infection in the absence of IFN-α but not as strongly as after IFN-α treatment. The inhibitory effect of IFN-α on *Ctr* was also mild. We included the results of this experiment as a new Figure 1J and added the following statement:

Line 189: Since IFN-α also signals through STAT1, we tested whether IFN-α also downregulates c-Myc and chlamydial growth. Although c-Myc levels and the bacterial load were slightly reduced upon IFN-a treatment, it failed to induce strong c-Myc depletion and chlamydial persistence suggesting that Type I and -II IFNs have different effects on c-Myc levels and chlamydial replication (Figure 1J).

(d) Consistency:There are differences in response to IFN γ between cell lines that are not addressed fully.Figure S3 shows c-Myc overexpression in HeLa cells but not in organoids, provide this result.

If we understand correctly, this comment addresses two issues: c-Myc response to IFN-γ in cell lines and the overexpression of c-Myc by lentiviral transduction. Indeed, c-Myc downregulation by IFN-γ treatment is differently affected in different cell lines. However, since c-Myc expression is weak in human organoids in the absence of infection, the downregulation is only visible in infected organoids. This is shown in Figure 2E. Regarding the lentivirus-induced overexpression, this is also weaker in organoids as compared to HeLa cells as shown in Figure 3 —figure supplement 1C and D. We have now indicated this fact in the manuscript:

Line 326:

“IFN-γ-induced c-Myc downregulation varies in different cell lines and in organoids downregulation is only visible in infected organoids due to weak c-Myc expression in uninfected human organoids (Figure 2E). Therefore, lentivirus-induced overexpression is also weaker in human organoids as compared to HeLa 229 cells as shown in Figure 3 —figure supplement 1C and D.”

Similarly, the degree to which DMKG and nucleosides affect persistence appears less consistent (in contrast to L-tryptophane) and possibly confounded by different cell lines. In Figure 6, the addition of DMKG to IFN-treated cells during infection rescues inclusion number to levels of the untreated control but significantly less in Figure 7D. The addition of nucleosides in the cells used in Figure 6 rescues persistence significantly less (as assessed by inclusion number) than DMKG, but there is no data presented from the cells used in Figure 7.

We are aware of the differences in the rescue effect of DMKG and nucleosides shown in these two figures. In Figure 7, we investigated the effect of the chlamydial trpBA mutant. This experiment was done by the student who generated the trpBA mutant with a different batch of cells and Chlamydia than the experiment performed by another person in Figure 6. We simply wanted to demonstrate in this experiment that mutant Chlamydia behave as wildtype and can be rescued with metabolites. Since the data are consistent in this set of experiment and confirm the previous observation of a rescue with metabolites, we decided to show it as a figure although the rescue did not reach the same efficiency as in Figure 6.

The organoid infection experiments seemed to be underpowered and quantification of infectivity is a little suspect (Figure S2H- inclusions were titers an uneven number of cells). U2OS and HeLa cells are transformed and their metabolism is likely to be very distinct. Thus, show if DMKG or nucleosides rescue persistence in primary Fimb cells.

We agree that quantification of infection effects is challenging in the organoid model. We therefore repeated the experiment in Fimb cells as suggested and show now a clear rescue by supplementing with DMKG. These data have been included as Figure 6H.

In addition, the reduction of LAT1 levels is quite striking in HeLa cells (Figure 5C) but very minor in U2OS cells. U2OS cells might survive better c-Myc overexpression that can be pro-apoptotic. Please acknowledge.

We discuss this possible connection in the revised version of the manuscript.

Line 468:

“Differences in reduction of LAT1 levels between HeLa 229 and U2OS cells might be due to a better survival of c-Myc overexpression that can be pro-apoptotic in U2OS cells.”

(e) In addition, the model linking IFNγ-induced tryptophan limitation to destabilisation of c-Myc via PI3K-GSK3b is not fully supported by the current evidence. Either provide direct evidence of either increased phosphorylation of threonine 58 on c-Myc or increased proteasomal degradation of c-Myc as a result of decreased levels of tryptophan upon IFNγ treatment (at the moment, it seems that phosphorylation of threonine 58 is unchanged upon IFNγ treatment (see Lines 157-162 and Figure 1E)); or remove the model and rework the text accordingly. Of note, bacterial replication also induces AKT and GSK3b.

It indeed appears that phosphorylation of threonine 58 is not affected by IFN-γ treatment, however, this strongly depends on the way we normalize the western blots. We in general normalize to actin (as loading control) to have similar normalization conditions in all figures. In case of such a strong variation of c-Myc levels like after IFN-g treatment, the normalization to loading controls gives a wrong picture. If we normalize pThr58 to c-Myc levels, we get a 4- and 7-fold increase after IFN-γ and IFN-γ/infection, respectively. These data are therefore in line with a role of pThr58 in the downregulation of c-Myc and the model we propose for GSK3b and AKT. The model we included in the manuscript shows more or less the textbook knowledge of IFN-g signalling, which we included for the better understanding of the readers. We are happy to remove these models if the reviewer prefer their deletion.

We have included the following statement in Line 168:

“while phosphorylation at threonine 58 was unchanged (Figure 1E). These results were obtained, if protein bands of the western blot were normalized to actin as loading control. However, since c-Myc levels varied significantly upon IFN-γ exposure, normalization to c-Myc protein levels revealed strongly increased phosphorylation at Thr58 (4- and 7-fold, IFN- γ and IFN- γ/infected, respectively) and a mild increase at Ser62 (1.3- and 2-fold, IFN- γ and IFN- γ/infected, respectively). These data are consistent with a role for Thr58 phosphorylation in IFN-γ-mediated downregulation of c-Myc.”

(f) Line 541-544/Figure S6E: Considering that the most pronounced effect appears to be on adenosine (Figure 6C) and ATP (Figure S6E), please consider and discuss that the effect could be mediated by reduced ATP levels generally slowing down metabolic processes rather than DNA replication in particular.

We added the following sentence in Line 626:

“Another possibility is that IFN-γ-induced reduction of ATP levels generally slowed down metabolic processes and induced chlamydial persistence.”

(g) Line 553-555/Figure 6F. The authors suggest that "restoration of chlamydial development could also be achieved by the sole addition of pyrimidine (U+C), and to a minor extend [sic] also purine nucleosides (A+G)". However, Figure 6F appear to suggest that purines (A+G) have a more significant contribution to *Chlamydia trachomatis* development than pyrimidines (C+U), which appear to have a statistically insignificant effect. Please clarify.

We thank the reviewers for this comment since this conclusion has to be explained better. If we talk about development, we usually test this in a secondary infection assay. This assay tells us how many of the bacteria developed to EB under certain conditions and can therefore produce progeny. This is the immunoblot in Figure 6F. In this assay of chlamydial development pyrimidines appear to rescue better. The other assay is quantification of inclusions in primary infection where the purines have a significant effect. Since inclusion formation and infectivity are different readouts, we clarified this now in the revised version of the manuscript.

Line 635: Moreover, restoration of chlamydial development could also be achieved by the sole addition of pyrimidine nucleosides (U+C), and to a minor extend also by purine nucleosides (A+G) (Western blot in Figure 6F). In contrast, purines seem to restore inclusion formation more efficiently than pyrimidines (bar diagram in Figure 6F).